

# On constraining the strength of the terrestrial CO$_2$ fertilization effect in an Earth system model

**V. K. Arora and J. F. Scinocca**

Canadian Centre for Climate Modelling and Analysis, Environment and Climate Change Canada, University of Victoria, Victoria, BC, V8W 2Y2, Canada

Received: 17 November 2015 – Accepted: 9 December 2015 – Published: 15 January 2016

Correspondence to: V. K. Arora (vivek.arora@ec.gc.ca)

Published by Copernicus Publications on behalf of the European Geosciences Union.

GMDD

doi:10.5194/gmd-2015-252

On constraining the strength of the terrestrial CO$_2$ fertilization effect

V. K. Arora and J. F. Scinocca

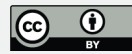

## Abstract

Earth system models (ESMs) explicitly simulate the interactions between the physical climate system components and biogeochemical cycles. Physical and biogeochemical aspects of ESMs are routinely compared against their observation-based counterparts to assess model performance and to evaluate how this performance is affected by ongoing model development. Here, we assess the performance of version 4.2 of the Canadian Earth system model against four, land carbon cycle focused, observation-based determinants of the global carbon cycle and the historical global carbon budget over the 1850–2005 period. Our objective is to constrain the strength of the terrestrial $CO_2$ fertilization effect which is known to be the most uncertain of all carbon cycle feedbacks. The observation-based determinants include (1) globally-averaged atmospheric $CO_2$ concentration, (2) cumulative atmosphere–land $CO_2$ flux, (3) atmosphere–land $CO_2$ flux for the decades of 1960s, 1970s, 1980s, 1990s and 2000s and (4) the amplitude of the globally-averaged annual $CO_2$ cycle and its increase over the 1980 to 2005 period. The optimal simulation that satisfies constraints imposed by the first three determinants yields a net primary productivity (NPP) increase from $\sim 58\,\mathrm{Pg\,C\,yr^{-1}}$ in 1850 to about $\sim 74\,\mathrm{Pg\,C\,yr^{-1}}$ in 2005; an increase of $\sim 27\,\%$ over the 1850–2005 period. The simulated loss in the global soil carbon amount due to anthropogenic land use change over the historical period is also broadly consistent with empirical estimates. Yet, it remains possible that these determinants of the global carbon cycle are insufficient to adequately constrain the historical carbon budget, and consequently the strength of terrestrial $CO_2$ fertilization effect as it is represented in the model, given the large uncertainty associated with LUC emissions over the historical period.

Discussion Paper | Discussion Paper | Discussion Paper | Discussion Paper |

**GMDD**

doi:10.5194/gmd-2015-252

**On constraining the strength of the terrestrial CO₂ fertilization effect**

V. K. Arora and
J. F. Scinocca

# 1 Introduction

The evolution of the atmospheric $CO_2$ concentration in response to anthropogenic fossil fuel $CO_2$ emissions is determined by the rate at which a fraction of these emissions is taken up by the land and ocean. Had the land and ocean not provided this "ecosystem service" since the start of the industrial era, and not removed about 50 % of $CO_2$ emissions from the atmosphere (Knorr, 2009), the present concentration of $CO_2$ in the atmosphere would have been around 500 ppm, compared to its current value of around 400 ppm. The manner in which the land and ocean will continue to provide this ecosystem service in future is of both scientific and policy relevance.

Future projections of atmospheric $CO_2$ concentration, $[CO_2]$, in response to continued anthropogenic $CO_2$ emissions, or alternatively projections of $CO_2$ emissions compatible with a given future $[CO_2]$ pathway, are based primarily on comprehensive Earth system models (ESMs) which include interactive land and ocean carbon cycle components (Jones et al., 2013). The land and ocean carbon cycle components in ESMs respond both to increases in $[CO_2]$ as well as the associated changes in climate. These carbon components also respond to changes in climate associated with other forcings including changes in concentration of non-$CO_2$ greenhouse gases and aerosols, to nitrogen deposition and over land to anthropogenic land use change (LUC).

The response of land and ocean carbon cycle components to changes in $[CO_2]$ and the associated change in climate is most simply characterized in the framework of the 140-year long 1 % year$^{-1}$ increasing $CO_2$ (1pctCO2) experiment, in which $[CO_2]$ increases at a rate of 1 % year$^{-1}$ from pre-industrial value of about 285 ppm until concentration quadruples to about 1140 ppm. The 1pctCO2 experiment has been recognized as a standard experiment by the coupled model intercomparison project (CMIP) which serves to quantify the response of several climate and Earth system metrics to increasing $CO_2$. These metrics include the transient climate response (TCR) and the transient climate response to cumulative emissions (TCRE, Gillett et al., 2013). Arora et al. (2013) analyzed results from fully-, biogeochemically- and radiatively-coupled

Discussion Paper | Discussion Paper | Discussion Paper | Discussion Paper | Discussion Paper |

**GMDD**

doi:10.5194/gmd-2015-252

**On constraining the strength of the terrestrial CO2 fertilization effect**

V. K. Arora and
J. F. Scinocca

versions of the 1pctCO2 experiment from eight ESMs that participated in the phase five of the CMIP (CMIP5). They calculated the response of land and ocean carbon cycle components to changes in [$CO_2$] and the associated change in climate expressed in terms of carbon-concentration and carbon–climate feedbacks, respectively. Arora et al. (2013) found that of all the carbon cycle feedbacks, the carbon-concentration feedback over land, which is primarily determined by the strength of the terrestrial $CO_2$ fertilization effect, is the most uncertain across models. They found that while the uncertainty in the carbon-concentration feedback over land had somewhat reduced since the first coupled carbon cycle climate model intercomparison project (C[4]MIP) (Friedlingstein et al., 2006) its uncertainty remained the largest of all carbon cycle feedbacks.

The reason for this large uncertainty is that it is fairly difficult at present to constrain the strength of the terrestrial $CO_2$ fertilization effect at the global scale. The net atmosphere–land $CO_2$ flux since the start of the industrial era has not only been influenced by the changes in [$CO_2$] but also the associated change in climate (due both to changes in [$CO_2$] and other climate forcers), nitrogen deposition, and more importantly land use change – the contribution of which itself remains highly uncertain. Since it is difficult to estimate the observed magnitude of net atmosphere–land $CO_2$ flux since the start of the industrial era attributable only to increase in [$CO_2$] it is consequently difficult to estimate the strength of the terrestrial $CO_2$ fertilization effect.

Measurements at Free-Air $CO_2$ Enrichment (FACE) sites in which vegetation is exposed to elevated levels of [$CO_2$] help to assess some aspects of $CO_2$ fertilization and how nutrients constraints regulate photosynthesis at elevated [$CO_2$] (Medlyn et al., 1999; McGuire et al., 1995). However, FACE results cannot be easily extrapolated to the global scale and the response of vegetation corresponds to a step increase in [$CO_2$] not the gradual increase which the real world vegetation is experiencing.

As part of the ongoing evaluation of carbon cycle in ESMs, the model simulated aspects of the global carbon cycle are routinely evaluated against their observation-based counterparts. These evaluations also provide the opportunity to adjust physical

Discussion Paper | Discussion Paper | Discussion Paper | Discussion Paper | Discussion Paper |

**GMDD**

doi:10.5194/gmd-2015-252

**On constraining the strength of the terrestrial CO₂ fertilization effect**

V. K. Arora and
J. F. Scinocca

Title Page

Abstract   Introduction

Conclusions   References

Tables   Figures

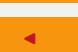   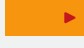

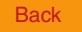   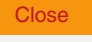

processes that influence the strength of the terrestrial $CO_2$ fertilization effect to provide the best comparison with observation-based aspects of the global carbon cycle. Here, we present results from such an evaluation for a new version of the Canadian Earth system model (CanESM4.2). An earlier version of the Canadian Earth system model (CanESM2, Arora et al., 2011) participated in the CMIP5 (Taylor et al., 2012) and its results also contributed to the fifth assessment report (AR5) of the Intergovernmental Panel on Climate Change (IPCC). We evaluate the response of CanESM4.2, for three different strengths of the terrestrial $CO_2$ fertilization effect, against four observation-based determinants of the global carbon cycle and the historical global carbon budget over the 1850–2005 period, with a focus on the land carbon cycle component. These determinants include (1) globally-averaged atmospheric $CO_2$ concentration, (2) cumulative atmosphere–land $CO_2$ flux, (3) atmosphere–land $CO_2$ flux for the decades of 1960s, 1970s, 1980s, 1990s and 2000s, and (4) the amplitude of the globally-averaged annual $CO_2$ cycle and its increase over the 1980 to 2005 period.

The strength of the $CO_2$ fertilization effect influences all four of these determinants of the global carbon cycle and the historical carbon budget. A stronger $CO_2$ fertilization effect, of course, implies a larger carbon uptake by land and consequently a lower rate of increase of [$CO_2$] in response to anthropogenic fossil fuel emissions. However, the strength of the $CO_2$ fertilization effect also influences the amplitude of the annual [$CO_2$] cycle which is primarily controlled by the Northern Hemisphere's biospheric activity. The amplitude of the annual [$CO_2$] cycle has been observed to increase over the past five decades suggesting a gradual increase in photosynthesis in association with a strengthening of the $CO_2$ fertilization effect (Keeling et al., 1996; Randerson et al., 1997) and thus possibly can help to constrain the strength of the terrestrial $CO_2$ fertilization effect in Earth system models.

**GMDD**

doi:10.5194/gmd-2015-252

**On constraining the strength of the terrestrial CO₂ fertilization effect**

V. K. Arora and J. F. Scinocca

GMDD

doi:10.5194/gmd-2015-252

**On constraining the strength of the terrestrial $CO_2$ fertilization effect**

V. K. Arora and
J. F. Scinocca

Interactive Discussion

## 2  The coupled climate–carbon system and CanESM4.2

### 2.1  The coupled climate–carbon system

The globally-averaged and vertically-integrated carbon budget for the combined atmosphere–land–ocean system may be written as:

$$\frac{dH_G}{dt} = \frac{dH_A}{dt} + \frac{dH_L}{dt} + \frac{dH_O}{dt} = E_F \tag{1}$$

where the *Global* carbon pool $H_G = H_A + H_L + H_O$ is the sum of carbon in the *Atmosphere*, *Land* and *Ocean* components, respectively (Pg C), and $E_F$ is the rate of anthropogenic fossil fuel $CO_2$ emissions (Pg C yr$^{-1}$) into the atmosphere. The equations for the atmosphere, land and ocean components are written as

$$\frac{dH_A}{dt} = F_A + E_F$$
$$= -F_L - F_O + E_F$$
$$= -(F_I - E_L) - F_O + E_F$$
$$= -F_I - F_O + E_F + E_L \tag{2}$$

$$\frac{dH_L}{dt} = F_L = F_I - E_L$$

$$\frac{dH_O}{dt} = F_O$$

where $(F_L + F_O) = -F_A$ are the fluxes (Pg C yr$^{-1}$) between the atmosphere and the underlying land and ocean, taken to be positive into the components. The net atmosphere–land $CO_2$ flux $F_L = F_I - E_L$ is composed of LUC emission rate $E_L$ (Pg C yr$^{-1}$) as well as the remaining global "natural" $CO_2$ flux $F_I$ that is often referred to as the residual or missing land sink in the context of the historical carbon budget

(Le Quéré et al., 2015). The emissions associated with LUC occur when natural vegetation, for example, is deforested and replaced by croplands resulting in net loss of carbon from land to the atmosphere (i.e. positive $E_L$). Conversely, when croplands are abandoned and gradually replaced by forests then carbon is gained from atmosphere into the land (i.e. negative $E_L$).

Over land, the rate of change of carbon is reflected in the model's three land pools (vegetation, V; soil, S; and litter or detritus, D)

$$
\begin{aligned}
\frac{dH_L}{dt} &= F_L = F_I - E_L \\
&= \frac{dH_V}{dt} + \frac{dH_S}{dt} + \frac{dH_D}{dt} \\
&= (G - R_A) - R_H - E_L \\
&= N - R_H - E_L
\end{aligned}
\tag{3}
$$

where $G$ is the gross primary productivity ($\mathrm{Pg\,C\,yr^{-1}}$) which represents the rate of carbon uptake by vegetation through photosynthesis, and $R_A$ and $R_H$ are the autotrophic and heterotrophic respiratory fluxes ($\mathrm{Pg\,C\,yr^{-1}}$) from living vegetation and dead litter and soil carbon pools, respectively. $N = G - R_A$ is the net primary productivity (NPP) which represents the carbon uptake by vegetation after autotrophic respiratory costs have been taken into account. The heterotrophic respiration $R_H = R_{H,D} + R_{H,S}$ is composed of respiration from the litter and soil carbon pools. The rate of change in carbon in model's litter ($H_D$) and soil ($H_S$) pools is written as

$$
\begin{aligned}
\frac{dH_D}{dt} &= D_L + D_S + D_R - C_{D \to S} - R_{H,D} \\
\frac{dH_S}{dt} &= C_{D \to S} - R_{H,S}
\end{aligned}
\tag{4}
$$

where $D_{i,\,i=L,S,R}$ is the litter fall from the model's Leaf, Stem and Root components into the model's litter pool. $C_{D \to S}$ is the transfer of humidified litter into the soil carbon pool

Discussion Paper | Discussion Paper | Discussion Paper | Discussion Paper |

**GMDD**

doi:10.5194/gmd-2015-252

**On constraining the strength of the terrestrial CO$_2$ fertilization effect**

V. K. Arora and J. F. Scinocca

calculated as a fraction of the litter respiration ($R_{H,D}$)

$$C_{D \to S} = \chi R_{H,D} \tag{5}$$

and $\chi$ is the humification factor.

Integrating Eqs. (2) and (3) in time with $\int_{t_0}^{t} (dH/dt)dt = H(t) - H(t_0) = \Delta H(t)$ and $\int_{t_0}^{t} F dt = \tilde{F}(t)$ (Pg C) gives

$$
\begin{aligned}
\Delta H_A &= -\left(\tilde{F}_O + \tilde{F}_I\right) + \left(\tilde{E}_F + \tilde{E}_L\right) \\
\Delta H_O &= \tilde{F}_O \\
\Delta H_L &= \tilde{F}_L = \tilde{F}_I - \tilde{E}_L; \\
&= \Delta H_V + \Delta H_S + \Delta H_D = \tilde{F}_I - \tilde{E}_L = \tilde{N} - \tilde{R}_H - \tilde{E}_L \\
\Delta H_I &= \tilde{F}_I \\
\Delta H &= \tilde{E}_F
\end{aligned}
\tag{6}
$$

The cumulative change in the atmosphere, the ocean and the land carbon pools is written as

$$
\begin{aligned}
\Delta H_A + \Delta H_O + \left(\Delta H_I - \tilde{E}_L\right) &= \tilde{E}_F \\
\Delta H_A + \Delta H_O + \Delta H_I &= \tilde{E}_F + \tilde{E}_L = \tilde{E}
\end{aligned}
\tag{7}
$$

where $\tilde{E}$ (Pg C) is the cumulative sum of the anthropogenic emissions from fossil fuel consumption and land use change. When emissions associated with LUC are zero, Eq. (7) becomes

$$\Delta H_A + \Delta H_O + \Delta H_L = \tilde{E}_F = \tilde{E} \tag{8}$$

which indicates how cumulative emissions are parsed into changes in atmospheric carbon burden and carbon uptake by the ocean and land components.

Discussion Paper | Discussion Paper | Discussion Paper | Discussion Paper | Discussion Paper |

**GMDD**

doi:10.5194/gmd-2015-252

**On constraining the strength of the terrestrial CO$_2$ fertilization effect**

V. K. Arora and J. F. Scinocca

## 2.2 Canadian Earth System Model version 4.2

### 2.2.1 Physical components

At the Canadian Centre for Climate Modelling and Analysis (CCCma), the earth system model, CanESM2, has undergone further development since its use for CMIP5. This version of the model has been equivalently labelled CanESM4.0 in an effort to rationalize the ESM naming convention to better reflect the fact that this model version employs the 4th generation atmosphere component, CanAM4, (Von Salzen et al., 2013) and the 4th generation ocean component, CanOM4 (Arora et al., 2011). The version of the CCCma earth system model used for this study is CanESM4.2 and so, represents two full cycles of model development on all of its components. Similar to CanESM2, the physical ocean component of CanESM4.2 (CanOM4.2) has 40 levels with approximately 10 m resolution in the upper ocean while the horizontal ocean resolution is approximately 1.41° (longitude) × 0.94° (latitude). The majority of development in CanESM4.2, relative to CanESM2, has occurred on its atmospheric component CanAM4.2. CanAM4.2 is a spectral model employing T63 triangular truncation with physical tendencies calculated on a 128 × 64 (∼ 2.81°) horizontal linear grid with 49 layers in the vertical whose thicknesses increase monotonically with height to 1 hPa. Relative to CanAM4, CanAM4.2 includes a new version of the Canadian Land Surface Scheme, CLASS3.6, which models the energy and water fluxes at the atmosphere–land boundary by tracking energy and water through the soil, snow, and vegetation canopy components (Verseghy, 2012). CLASS models the land surface energy and water balance and calculates liquid and frozen soil moisture, and soil temperature for three soil layers (with thicknesses 0.1, 0.25 and 3.75 m). The thickness of the third layer depends on the depth to bedrock (and is in many places less than 3.75 m) based on the Zobler (1986) soil data set. Changes to CLASS primarily include improvements to the simulation of snow at the land surface. These incorporate new formulations for vegetation interception of snow (Bartlett et al., 2006), for unloading of snow from vegetation (Hedstrom and Pomeroy, 1998), for the albedo of snow-covered canopies

Discussion Paper | Discussion Paper | Discussion Paper | Discussion Paper | Discussion Paper |

**GMDD**

doi:10.5194/gmd-2015-252

**On constraining the strength of the terrestrial CO2 fertilization effect**

V. K. Arora and
J. F. Scinocca

(Bartlett and Verseghy, 2015), for limiting snow density as a function of depth (Tabler et al., 1990; Brown et al., 2006), and for the thermal conductivity of snow (Sturm et al., 1997). Water retention in snowpacks has also been incorporated. CanAM4.2 also includes an aerosol microphysics scheme (von Salzen, 2006; Ma et al., 2008; Peng et al., 2012), a higher vertical resolution in the upper troposphere, a reduced solar constant (1361 W m$^{-2}$) and an improved treatment of the solar continuum used in the radiative transfer. CanAM4.2 also considers natural and anthropogenic aerosols and their emissions, transport, gas-phase and aqueous-phase chemistry, and dry and wet deposition as summarized in Namazi et al. (2015).

### 2.2.2 Land and ocean carbon cycle components

The ocean and land carbon cycle components of CanESM4.2, are similar to CanESM2, and represented by the Canadian Model of Ocean Carbon (CMOC) (Christian et al., 2010) and the Canadian Terrestrial Ecosystem Model (CTEM) (Arora et al., 2009; Arora and Boer, 2010), respectively.

LUC emissions in CTEM are modelled interactively on the basis of changes in land cover which are determined by changes in crop area. The historical land cover used in the simulations presented here is reconstructed using the linear approach of Arora and Boer (2010) and is the same as used for CMIP5 simulations; as the fraction of crop area in a grid cell changes, the fraction of non-crop plant functional types (PFTs) is adjusted linearly in proportion to their existing coverage. The historical changes in crop area are based on the data set provided for CMIP5 simulations as explained in Arora and Boer (2014). When the fraction of crop area in a grid cell increases then the fractional coverage of other PFTs is reduced which results in deforested biomass. The deforested biomass is allocated to three components that are (i) burned instantaneously and contribute to (ii) short (paper) and (iii) long (wood products) term pools (Arora and Boer, 2010). When the fraction of crop area decreases, the fractional coverage of non-crop PFTs increases and their vegetation biomass is spread over a larger area reducing vegetation density. Carbon is sequestered until a new equilibrium is reached providing

Discussion Paper | Discussion Paper | Discussion Paper | Discussion Paper | Discussion Paper |

**GMDD**

doi:10.5194/gmd-2015-252

**On constraining the strength of the terrestrial CO$_2$ fertilization effect**

V. K. Arora and J. F. Scinocca

a carbon sink associated with regrowth as the abandoned areas revert back to natural vegetation.

The LUC emissions term ($E_L$) in the Eqs. (1) through (8) is not easily defined or calculated. Pongratz et al. (2014) discuss the multiple definitions and methods of calculating $E_L$. When $E_L$ is calculated using models, it is most usually defined as the difference in $F_L$ between simulations with and without LUC. This is also the basic definition used by Pongratz et al. (2014). Calculating $E_L$ thus requires performing additional simulations without land use change in which land cover is held constant at its pre-industrial state. For a simulation without LUC Eq. (3) becomes

$$\frac{dH'_L}{dt} = F'_L = F'_I \tag{9}$$

and an estimate of $E_L$, and its cumulative values $\tilde{E}_L$, is obtained as

$$\begin{aligned} E_L &= F'_L - F_L \\ \tilde{E}_L &= \tilde{F}'_L - \tilde{F}_L \end{aligned} \tag{10}$$

Over the historical period, globally, $F'_L$ is expected to be higher than $F_L$ (both considered positive downwards) due, at least, to two processes: (1) fraction of deforested biomass that is burned and which contributes to short and long term product pools all release carbon to the atmosphere, albeit at different time scales, (2) the area that is deforested and put under agricultural use loses soil carbon and cannot sequester carbon in response to increase [$CO_2$] since crops are frequently harvested. As a result $E_L$ is positive.

Relative to CanESM2, the version of CTEM employed in CanESM4.2, CTEM4.2, includes changes to the humification factor ($\chi$, see Eqs. 4 and 5) which determines what fraction of the humidified litter is transferred from litter ($H_D$) to the soil carbon pool ($H_S$). The value of $\chi$ employed in CTEM4.2 has been changed for crop PFTs from 0.45 to 0.10, which decreases soil carbon when natural vegetation is converted

Discussion Paper | Discussion Paper | Discussion Paper | Discussion Paper |

**GMDD**

doi:10.5194/gmd-2015-252

**On constraining the strength of the terrestrial CO2 fertilization effect**

V. K. Arora and J. F. Scinocca

to croplands. As a result, a decrease in global soil carbon over the historical period is obtained as natural vegetation is replaced by croplands as would be expected based on empirical measurements (Wei et al., 2014). This change in humification factor was required despite the higher litter decomposition rates over croplands and is discussed in more detail later in the results section. In addition, in CTEM4.2 the sensitivity of photosynthesis to soil moisture is reduced for coupling to CLASS 3.6, especially for the broadleaf evergreen PFT (which exists mainly in the tropics) to somewhat account for deep roots, for example, in the Amazonian region (e.g. see da Rocha et al., 2004).

CTEM has always included a parameterization of photosynthesis down-regulation, which represents acclimatization to elevated $CO_2$ in the form of a decline in maximum photosynthetic rate. In the absence of explicit coupling of terrestrial carbon and nitrogen cycles this parameterization yields a mechanism to reduce photosynthesis rates as $[CO_2]$ increases. The photosynthesis down-regulation parameterization is described in detail in Arora et al. (2009). Briefly, the modelled "potential" gross photosynthesis rate ($G_p$), which is not constrained by nutrient limitation, is multiplied by a scalar $\xi(C)$ (Eq. 11) which yields the gross primary productivity ($G$) used in Eq. (3).

$$G = \xi(C)G_p$$

$$\xi(C) = \frac{1 + \gamma_d \ln(C/C_0)}{1 + \gamma_p \ln(C/C_0)} \tag{11}$$

where $\gamma_d < \gamma_p$. A lower value of $\gamma_d$ than $\gamma_p$ yields a value of $\xi(C)$ that is less than one. As the concentration of $CO_2$, expressed as $C$ in Eq. (11), increases above its pre-industrial level $C_0$ (285 ppm), $\xi(C)$ progressively decreases resulting in a gross primary productivity G, which is less than the its potential value $G_p$. Figure 1 shows the behaviour of $\xi(C)$ for $\gamma_p = 0.95$ and three values of $\gamma_d$ (0.25, 0.4 and 0.55) corresponding to three different strengths of the terrestrial $CO_2$ fertilization effect. The value of $\gamma_d = 0.25$ was used for CanESM2. Through the parameter $\gamma_d$, the physical process of down-regulation has a direct influence on the strength of the terrestrial $CO_2$ fertilization effect. In practice, different combinations of $\gamma_d$ and $\gamma_p$ are able to yield very similar

## GMDD

doi:10.5194/gmd-2015-252

**On constraining the strength of the terrestrial CO2 fertilization effect**

V. K. Arora and
J. F. Scinocca



values of $\xi(C)$. Arora et al. (2009) calculated the value of $\gamma_d$ based on results from six studies, two of which were meta-analyses each based on 15 and 77 individual studies, that grow plants in ambient and elevated $CO_2$ environment. Their results are equivalent to $\gamma_d = 0.46$ with a range from 0.22 to 0.63 for $\gamma_p = 0.95$.

### 2.2.3 Treatment of $CO_2$ in the atmosphere

The land and ocean components of the carbon cycle in CanESM4.2 are operable for two experimental designs – (1) the emissions-driven mode, where the atmospheric $CO_2$ concentration is a freely evolving 3-D tracer in the model and (2) concentrations-driven mode, where the atmospheric $CO_2$ concentration is prescribed externally.

In the emissions-driven mode the anthropogenic $CO_2$ emissions ($E_F$) are specified and since the interactive land and ocean carbon cycle components simulate the $F_L$ and $F_O$ terms, respectively, the model is able to simulate the evolution of $[CO_2]$ through the $H_A$ term, which represents the atmospheric carbon burden, in Eq. (2). This is referred to as the "free" or interactively simulated $[CO_2]$. In this case, the model simulates the transport of $CO_2$ in the atmosphere and as a result its 3-D structure in space, its annual cycle through a year and its inter-annual variability.

In the concentrations-driven mode, the land and ocean $CO_2$ fluxes, $F_L$ and $F_O$, remain interactively determined so model results can be used to diagnose the $E_F$ term (based on Eq. 2) that is compatible with a given $[CO_2]$ pathway at the global scale. The concentrations-driven mode can be executed in two CanESM4.2 configurations. In the first configuration, a single scalar value of $[CO_2]$ which may be time evolving, is imposed at all geographical and vertical locations in the model. This follows the CMIP5 prescription for concentrations-driven simulations and we refer to it here as, "specified-$CO_2$" concentrations-driven mode. In the second configuration, a new approach for specifying $CO_2$ concentration has been implemented in CanESM4.2. In this new approach, only the globally averaged concentration of $CO_2$ in the lowest model level is constrained by the prescribed value. The geographical and vertical distribution of $CO_2$ in the atmosphere and its annual cycle in this second configuration is essentially iden-

**[GMDD](doi:10.5194/gmd-2015-252)**

doi:10.5194/gmd-2015-252

**On constraining the strength of the terrestrial CO₂ fertilization effect**

V. K. Arora and J. F. Scinocca

tical to the emissions-driven, free-$CO_2$, mode except that it employs zero emissions and a strong relaxation on the global-mean value of [$CO_2$] in the lowest model level towards the specified reference value. A relaxation timescale of one day is employed in this configuration. The reference value of [$CO_2$] may be time evolving and includes a fixed annual cycle derived from the free-$CO_2$ preindustrial control simulation. We refer to this configuration as the "relaxed-$CO_2$" concentrations-driven mode.

There are a number of advantages to using the relaxed-$CO_2$ configuration over the specified-$CO_2$ configuration for concentrations-driven experiments. The relaxed-$CO_2$ configuration preserves the spatial structure and annual cycle of [$CO_2$] expressed in the fully free-CO2 simulations of CanESM4.2 while still presenting a prescribed [$CO_2$] to the land and ocean components of the carbon cycle. Additionally, when spinning up land and ocean carbon pools for a prescribed atmospheric $CO_2$ concentration in the preindustrial control simulation, the equilibrated state of the relaxed-$CO_2$ configuration is found to produce little or no drift when used to initialize the free-CO2 preindustrial control simulations. In fact, the relaxed-$CO_2$ preindustrial control simulation may be used as the control simulation for both emissions-driven and concentrations-driven experiments. This is not the case when the specified $CO_2$ configuration is employed.

## 3 Experimental set up

Three different kinds of experiments are performed for this study. The first is the standard 1 % year$^{-1}$ increasing $CO_2$ experiment (1pctCO2) performed for three different strengths of the terrestrial $CO_2$ fertilization effect. The 1pctCO2 is a concentration-driven experiment and we use the "relaxed-$CO_2$" configuration to specify $CO_2$ in the atmosphere. The second experiment is the CMIP5 1850–2005 historical experiment, referred to as esmhistorical following CMIP5 terminology, which is performed with specified anthropogenic $CO_2$ emissions (i.e. in emissions-driven, or "free-$CO_2$", mode), where [$CO_2$] is simulated interactively. Concentrations of non-$CO_2$ greenhouse gases and emissions of aerosols and their precursors are specified in the esmhistorical ex-

**GMDD**

doi:10.5194/gmd-2015-252

**On constraining the strength of the terrestrial CO2 fertilization effect**

V. K. Arora and
J. F. Scinocca

Discussion Paper | Discussion Paper | Discussion Paper | Discussion Paper | Discussion Paper

peri­ment following the CMIP5 protocol. The third experiment is same as the esmhis­torical experiment but LUC is not permitted and the land cover remains at its 1850 value; referred to as the esmhistorical_noluc experiment. Two ensemble members are performed for each of the three versions of the esmhistorical and esmhistorical_noluc

experiments corresponding to three different strengths of the terrestrial $CO_2$ fertiliza­tion effect. The rationale for performing historical simulations without LUC is to be able to quantify LUC emissions $E_L$ using Eq. (10). Table 1 summarizes all the simulations performed.

The 1pctCO2 simulations with "relaxed" $CO_2$ for three different strengths of the ter­
restrial $CO_2$ fertilization effect are initialized from a corresponding pre-industrial control simulation with $CO_2$ specified at $\sim 285$ ppm and all other forcings at their 1850 val­ues. The esmhistorical and esmhistorical_noluc simulations are initialized from a pre-industrial control simulation with "free" $CO_2$ and zero anthropogenic $CO_2$ emissions.

## 4   Results

### 4.1   1 % year$^{-1}$ increasing $CO_2$ experiments

Figure 2 shows the carbon budget components of Eq. (8); $\Delta H_A$, $\Delta H_O$ and $\Delta H_L$ i.e. the change in atmospheric carbon burden and cumulative atmosphere–ocean and atmosphere–land $CO_2$ flux which together make up the cumulative diagnosed emis­sions ($\tilde{E}$) based on results from the fully-coupled 1pctCO2 experiment. Results are
shown from eight CMIP5 models that participated in the Arora et al. (2013) study, including CanESM2 which used $\gamma_d = 0.25$, together with those from CanESM4.2 for three different strengths of the terrestrial $CO_2$ fertilization effect. The cumulative atmosphere–land $CO_2$ flux across models varies much more than the cumulative atmosphere–ocean $CO_2$ flux across the CMIP5 models as already noted in Arora
et al. (2013). The results for CanESM4.2 indicate that the influence of $\gamma_d$ (Eq. 11) on the strength of the model's terrestrial $CO_2$ fertilization effect allows CanESM4.2's cumu-

**GMDD**

doi:10.5194/gmd-2015-252

**On constraining the strength of the terrestrial CO$_2$ fertilization effect**

V. K. Arora and
J. F. Scinocca

Discussion Paper | Discussion Paper | Discussion Paper | Discussion Paper

**GMDD**

doi:10.5194/gmd-2015-252

lative diagnosed emissions to essentially span the range of the other CMIP5 models. For the three different strengths of the terrestrial $CO_2$ fertilization effect, $\gamma_d$ = 0.25, 0.4 and 0.55, the $\gamma_d$ values of 0.4 and 0.55 yield cumulative atmosphere–land $CO_2$ flux that is higher than all the CMIP5 models. The basis for choosing these values of $\gamma_d$ will become obvious later.

   The cumulative atmosphere–land $CO_2$ flux $\Delta H_L$ for CanESM4.2 for the simulation with $\gamma_d$ = 0.25 is higher than that for CanESM2 which also uses $\gamma_d$ = 0.25, because of the changes made to soil moisture sensitivity of photosynthesis and because $\Delta H_L$ also depends on the model climate. In particular, the CanESM2 bias of low precipitation over the Amazonian region has been reduced in CanESM4.2, as shown in Fig. 3. The increased precipitation over the Amazonian region causes increased carbon uptake with increasing [$CO_2$]. The improved precipitation bias of CanESM4.2 in this region is in part caused by the decreased sensitivity of photosynthesis to soil moisture in CTEM4.2, especially for broadleaf evergreen PFT, which helps to increase evapotranspiration and in turn increase precipitation over the region.

## 4.2   Historical simulations with LUC

The results presented in this section evaluate the model against four observation-based determinants of the global carbon cycle and the historical global carbon budget over the 1850–2005 period mentioned earlier. Simulated atmosphere–ocean $CO_2$ fluxes are also compared with observation-based estimates although, of course, they are not directly affected by the strength of the terrestrial $CO_2$ fertilization effect.

### 4.2.1   Components of land carbon budget

In Fig. 4, time series of instantaneous ($F_L$, panel a) and cumulative ($\tilde{F}_L$, panel b) atmosphere–land $CO_2$ flux over the period 1850–2005 are displayed for CanESM2 (which contributed results to CMIP5) and CanESM4.2 for the three different strengths of the terrestrial $CO_2$ fertilization effect. The observation-based estimates of $F_L = (F_I - E_L)$

**On constraining the strength of the terrestrial CO$_2$ fertilization effect**

V. K. Arora and J. F. Scinocca

in Fig. 4a for the decades of 1960, 1970, 1980, 1990 and 2000 are reproduced from Le Quéré et al. (2015) who derive the $F_L = (F_l - E_L)$ term as residual of the carbon budget equation $dH_A/dt = -(F_l - E_L) - F_O + E_F$ using observation-based estimates of change in atmospheric carbon budget ($dH_A/dt$), atmosphere–ocean $CO_2$ flux ($F_O$) and fossil

fuel emissions ($E_F$). The observation-based estimate of $-11 \pm 47\,\text{Pg C}$ in Fig. 4b for $\tilde{F}_L$ over the period 1850–2005 is from Arora et al. (2011) (their Table 1).

The primary difference between CanESM2 and CanESM4.2 simulations in Fig. 4 is that $\tilde{F}_L$ for CanESM2 generally stays positive throughout the historical period, whereas for CanESM4.2 it first becomes negative (indicating that land is losing carbon) and

10 then becomes positive (indicating that land is gaining carbon) towards the end of the 20th century, depending on the strength of the $CO_2$ fertilization effect. The behaviour of $\tilde{F}_L$ for CanESM4.2 is considered to be more realistic. As the land responds to anthropogenic land use change, associated with an increase in crop area early in the historical period, it causes a decrease in vegetation and soil carbon (see Fig. 5). Later

in the 20th Century, the $CO_2$ fertilization effect causes the land to become a sink for carbon resulting in both vegetation and soil carbon increases. This behavior is consistent with the mean model response of the 15 CMIP5 models analyzed by Hoffman et al. (2013) (their Fig. 2b). In contrast, CanESM2 shows a gradual increase in the global soil carbon amount (Fig. 5a) over the historical period. In Fig. 5, it can be seen

that the effect of $CO_2$ fertilization in the second half of the 20th century is delayed for soil carbon compared to that for vegetation. This is primarily because of the lag introduced by the turnover time of vegetation (i.e., increased NPP inputs have to go through vegetation pool first) and the longer turnover time scale of the soil carbon pool. The more reasonable response of soil carbon to anthropogenic land use change, in

Fig. 5a for CanESM4.2, is achieved by changing the humification factor from 0.45 (in CanESM2) to 0.10 (in CanESM4.2) in Eq. (5) which yields a reduction in global soil carbon amount in response to land use change up until the time that the effect of $CO_2$ fertilization starts to take effect. In Fig. 4a, CanESM4.2 is also able to simulate continuously increasing $F_L$ during the period 1960 to 2005, depending on the strength of

**GMDD**

doi:10.5194/gmd-2015-252

**On constraining the strength of the terrestrial CO$_2$ fertilization effect**

V. K. Arora and J. F. Scinocca

the $CO_2$ fertilization effect, while CanESM2 simulates near constant or decreasing $F_L$ from about 1990 onwards, as is also seen in Fig. 4b for $\tilde{F}_L$. This behaviour of $F_L$ is not consistent with observation-based estimates from Le Quéré et al. (2015) which show continued strengthening of the land carbon sink since 1960s.

In Fig. 4a, amongst the three versions of the CanESM4.2, the simulation with $\gamma_d = 0.4$ (blue line) yields the best comparison with observation-based estimates of $F_L$ from Le Quéré et al. (2015), while the simulations with $\gamma_d = 0.25$ (green line) and $\gamma_d = 0.55$ (red line) yield $F_L$ values that are lower and higher, respectively, than observation-based estimates. In Fig. 4b, the cumulative atmosphere–land $CO_2$ flux $\tilde{F}_L$ over the 1850–2005

period from the simulations with $\gamma_d = 0.25$ and 0.4 (green and blue lines, respectively) lies within the uncertainty of observation-based estimates, while the simulation with $\gamma_d = 0.55$ (red line) yields $\tilde{F}_L$ value that is high relative to observation-based estimate.

Figure 6 shows the change in and absolute values of NPP from CanESM2 and the simulations made with CanESM4.2 for three different strengths of the $CO_2$ fertilization

effect. Consistent with 1pctCO2 simulations, the rate of increase of NPP in CanESM4.2 with $\gamma_d = 0.25$ is higher than that in CanESM2 which also uses $\gamma_d = 0.25$. This is because the underlying model climate is different in CanESM2 and CanESM4.2, as mentioned earlier, and the fact that photosynthesis sensitivity to soil moisture has also been reduced. The rates of increase of NPP for $\gamma_d = 0.40$ and 0.50 are, of course,

even higher. The CanESM4.2 simulation with $\gamma_d = 0.40$, which yields the best comparison with observation-based estimates of $F_L$ for the decade of 1960 through 2000 (Fig. 4a) as well as $\tilde{F}_L$ for the period 1850–2005 (Fig. 4b), yields an increase in NPP of $\sim 16\,\mathrm{Pg\,C\,yr^{-1}}$ over the 1850–2005 period. A caveat here is that part of this increase is also caused by increase in the crop area over the historical period that is realized

in the model regardless of the strength of the $CO_2$ fertilization effect. In CTEM4.2, the maximum photosynthetic capacity of crops is higher than for other PFTs to account for the fact that agricultural areas are generally fertilized. As a result, increase in crop area also increases global NPP. The increasing crop productivity has been suggested to contribute to the increase in amplitude of the annual [$CO_2$] cycle since 1960s (Zeng

Discussion Paper | Discussion Paper | Discussion Paper | Discussion Paper | Discussion Paper |

**GMDD**

doi:10.5194/gmd-2015-252

**On constraining the strength of the terrestrial $CO_2$ fertilization effect**

V. K. Arora and
J. F. Scinocca

et al., 2014). However, in the absence of an explicit representation of terrestrial N cycle (and thus fertilization of cropped areas) or a representation of increase in crop yield per unit area due to genetic modifications, the only processes in CTEM that contribute to changes in crop yield are the change in crop area itself and the increase in crop NPP
due to the $CO_2$ fertilization effect.

### 4.2.2  Globally-averaged [$CO_2$]

Figure 7 shows the simulated globally-averaged surface [$CO_2$] from the emissions-driven esmhistorical simulation of CanESM2 and that of CanESM4.2 for three different strengths of the $CO_2$ fertilization effect. The observation-based time series of
[$CO_2$] is illustrated by the heavy black line. The CanESM2 ($\gamma_d$ = 0.25) simulation yields a reasonable comparison with observation-based [$CO_2$]. Amongst the versions of CanESM4.2 with different strengths of the $CO_2$ fertilization effect, the version with $\gamma_d$ = 0.40 yields the best comparison. The CanESM4.2 version with $\gamma_d$ = 0.25 (weaker strength of the $CO_2$ fertilization effect) and 0.55 (stronger $CO_2$ fertilization effect) yield
$CO_2$ concentrations that are respectively higher and lower than the observational estimate from roughly mid-20th Century onward. The reason CanESM4.2 ($\gamma_d$ = 0.40) requires a stronger $CO_2$ fertilization effect than CanESM2 ($\gamma_d$ = 0.25) for simulating the observation-based increase in atmospheric $CO_2$ burden over the historical period is the enhanced impact of LUC in CanESM4.2 due to its increased humification factor
and the associated response of the global soil carbon pool, as discussed in the previous section. The differences in simulated [$CO_2$] in Fig. 7 from CanESM4.2 are due only to differences in the strength of the $CO_2$ fertilization effect. Although, of course, since in these simulations [$CO_2$] is simulated interactively, the simulated atmosphere–land flux $F_L$ and [$CO_2$] both respond to and affect each other.
Both CanESM2 and CanESM4.2 under predict [$CO_2$] relative to observational estimates over the period 1850–1930, and are also unable to reproduce the near zero rate of increase of [$CO_2$] around 1940. Possible reasons for these discrepancies include (1) the possibility that carbon cycle before 1850 was not in true equilibrium and

**GMDD**

doi:10.5194/gmd-2015-252

**On constraining the strength of the terrestrial CO$_2$ fertilization effect**

V. K. Arora and J. F. Scinocca

this aspect cannot be captured since the model is spun up to equilibrium for 1850 conditions, (2) the uncertainties associated with anthropogenic emissions for the late 19th and early 20th century that are used to drive the model, and (3) the uncertainties associated with pre Mauna-Loa [$CO_2$] observations.

### 4.2.3 Atmosphere–ocean CO2 flux

Figure 8a and b, respectively, show time series of instantaneous ($F_O$) and cumulative ($\tilde{F}_O$) atmosphere–ocean $CO_2$ fluxes over the period 1850–2005 for the set of emissions-driven simulations presented in Fig. 7. The strength of the terrestrial $CO_2$ fertilization effect has little or no impact on the ocean biogeochemical processes. The difference in values of $F_O$ and $\tilde{F}_O$ for the three versions CanESM4.2 are, therefore, primarily due to the differences in [$CO_2$]. The observation-based estimates of $F_O$ in Fig. 8a for the decades of 1960, 1970, 1980, 1990 and 2000 are from Le Quéré et al. (2015). The observation-based estimate of $\tilde{F}_O$ of $141 \pm 27$ Pg C in Fig. 8b for the period 1850–2005 is from Arora et al. (2011) (their Table 1).

Both CanESM2 and the CanESM4.2 simulation for $\gamma_d = 0.40$ (which provides the best comparison with observation-based estimate for [$CO_2$]; blue line in Fig. 7) yield lower $\tilde{F}_O$ compared to observation-based values. The $F_O$ value from CanESM2 and the CanESM4.2 simulation for $\gamma_d = 0.40$ are lower than the mean estimates from Le Quéré et al. (2015) for the decades of 1960s through 2000s, although still within their uncertainty range. The family of ESMs from CCCma, all of which have the same physical ocean model, including CanESM1 (Arora et al., 2009), CanESM2 (Arora et al., 2011) and now CanESM4.2, yield lower than observed ocean carbon uptake over the historical period. Recent analyses of these model versions suggest that the primary reason for their low carbon uptake is a negative bias in near surface wind speeds over the Southern Ocean and an iron limitation in the same region which is too strong (N. Swart, personal communication, 2015, Canadian Centre for Climate Modelling and Analysis). The CanESM4.2 simulation with $\gamma_d = 0.25$ (green line in Fig. 8) yields a better comparison with observation-based estimates of $F_O$ and $\tilde{F}_O$ but that is because of

Discussion Paper | Discussion Paper | Discussion Paper | Discussion Paper | Discussion Paper

**GMDD**

doi:10.5194/gmd-2015-252

**On constraining the strength of the terrestrial CO2 fertilization effect**

V. K. Arora and J. F. Scinocca

the higher simulated $[CO_2]$ in that simulation associated with lower carbon uptake by land.

### 4.2.4 Amplitude of the annual $CO_2$ cycle

The annual $CO_2$ cycle is influenced strongly by the terrestrial biospheric activity of the
5 Northern Hemisphere (Keeling et al., 1996; Randerson et al., 1997). Higher than nor-
mal biospheric uptake of carbon during a Northern Hemisphere's growing season, for
example, will yield lower than normal $[CO_2]$ by the end of the growing season, around
September when $[CO_2]$ is at its lowest level (see Fig. 9a). Similarly, during the Northern
Hemisphere's dormant season, increased respiration from live vegetation and decom-
10 position of dead carbon, including leaf litter, that may be associated with increased
carbon uptake during the last growing season, will yield higher than normal $[CO_2]$ dur-
ing April when $[CO_2]$ is at its highest level. Both processes increase the amplitude of
the annual $[CO_2]$ cycle. Given this strong control, the rate of change of the amplitude
of the annual $[CO_2]$ cycle can potentially help to constrain the strength of the terrestrial
$CO_2$ fertilization effect.

Figure 9a compares the annual cycle of the trend-adjusted globally-averaged near-
surface monthly $[CO_2]$ anomalies from CanESM2 and the versions of CanESM4.2 for
three different strengths of the $CO_2$ fertilization effect with observation-based estimates
for the 1991–2000 period. Figure 9b shows the time series of the amplitude of the an-
20 nual cycle of the trend adjusted globally-averaged near-surface monthly $[CO_2]$ anoma-
lies (referred to as $\Phi_{CO_2}$) from CanESM2 and CanEM4.2, as well as observation-
based estimates going back to 1980s. While $CO_2$ measurements at Mauna Loa started
in 1959, observation-based globally-averaged near-surface $[CO_2]$ values are only
available since 1980s (ftp://aftp.cmdl.noaa.gov/products/trends/co2/co2_mm_gl.txt). In
Fig. 9b, consistent with the strengthening of the $CO_2$ fertilization effect, associated with
the increase in $[CO_2]$, the observation-based estimate of $\Phi_{CO_2}$ shows an increase
from 1980s to the present. Both CanESM2 and versions of CanESM4.2 also show an
increase in the amplitude of $\Phi_{CO_2}$ over the period 1850–2005. However, the absolute

Discussion Paper | Discussion Paper | Discussion Paper | Discussion Paper | Discussion Paper |

**GMDD**

doi:10.5194/gmd-2015-252

**On constraining the strength of the terrestrial CO2 fertilization effect**

V. K. Arora and
J. F. Scinocca

values of $\Phi_{CO_2}$ are lower in CanESM2 than in CanESM4.2 (Fig. 9b). Of course, in the absence of an observation-based estimate of pre-industrial value of $\Phi_{CO_2}$ it is difficult to say which value is more correct. However, when considering the present day values of $\Phi_{CO_2}$ the three versions of CanESM4.2 yield better comparison with observation-based estimate as also shown in Fig. 9a. The increase in the value of $\Phi_{CO_2}$ from CanESM2 to CanESM4.2, which now yields better comparison with observation-based value of $\Phi_{CO_2}$, is most likely caused by the change in the land surface scheme from CLASS 2.7 (that is implemented in CanESM2) to CLASS 3.6 (implemented in CanESM4.2), since the atmospheric component of the model hasn't changed substantially. It is, however, difficult to attribute the cause of this improvement in the present day value of $\Phi_{CO_2}$ in CanESM4.2 to a particular aspect of the new version of the land surface scheme. The annual [$CO_2$] cycle is driven primarily by the response of the terrestrial biosphere to the annual cycle of temperature and the associated greening of the biosphere every summer in the Northern Hemisphere. However, the simulated amplitude of the annual cycle of near-surface temperature hasn't changed substantially from CanESM2 to CanESM4.2 (not shown).

In Fig. 9b, the simulated values of $\Phi_{CO_2}$ for the CanESM4.2 simulations with $\gamma_d = 0.25$, 0.40 and 0.55 are 4.41, 4.69 and 4.85 ppm, respectively, averaged over the period 1991–2000, compared to observation-based value of $\Phi_{CO_2}$ of 4.36 ppm. Here, CanESM4.2 simulation with $\gamma_d = 0.25$ yields the best comparison with observation-based value of $\Phi_{CO_2}$. An increase in the strength of the $CO_2$ fertilization effect increases the amplitude of the annual [$CO_2$] cycle so a larger value of $\gamma_d$ yields a larger value of $\Phi_{CO_2}$. The increase in the amplitude of the annual [$CO_2$] cycle comes both from lower [$CO_2$] at the end of the growing season in September as well as higher [$CO_2$] at the start of the Northern Hemisphere's growing season in April (see Fig. 9a), as mentioned earlier in this section.

More important than the absolute value of $\Phi_{CO_2}$ is its rate of increase over time which is a measure of the strength of the terrestrial $CO_2$ fertilization effect. Figure 9b also shows the trend in $\Phi_{CO_2}$ over the 1980–2005 overlapping period for which both the

**GMDD**

doi:10.5194/gmd-2015-252

**On constraining the strength of the terrestrial CO2 fertilization effect**

V. K. Arora and
J. F. Scinocca

model and observation-based estimates of $\Phi_{CO_2}$ are available. The magnitude of trend for observation-based estimate of $\Phi_{CO_2}$ is $0.142 \pm 0.08 \, \text{ppm} \, (10 \, \text{yr})^{-1}$ (mean ± standard deviation, $\overline{x} \pm \sigma_x$), implying that over the 26 year 1980–2005 period the amplitude of annual [$CO_2$] cycle has increased by $0.37 \pm 0.21 \, \text{ppm}$. The calculated mean and stan-
dard deviation of the observation-based trend, however, does not take into account the uncertainty associated with the observation-based estimates of [$CO_2$], consideration of which will increase the calculated standard deviation even more. The magnitudes of trend in $\Phi_{CO_2}$ simulated by CanESM2 ($\gamma_d = 0.25$) and CanESM4.2 (for $\gamma_d = 0.25$) are $0.103 \pm 0.05$ and $0.153 \pm 0.031 \, \text{ppm} \, (10 \, \text{yr})^{-1}$, respectively, and statistically not dif-
ferent from the trend in the observation-based value of $\Phi_{CO_2}$ implying an increase of $0.27 \pm 0.13$ and $0.40 \pm 0.08 \, \text{ppm}$, respectively, in $\Phi_{CO_2}$ over the 1980–2005 period. The statistical difference is calculated on the basis of $\overline{x} \pm 1.385 \, \sigma_x$ range which corresponds to 83.4 % confidence intervals; the estimates from two sources are statistically not dif-
ferent at the 95 % confidence level if this range overlaps (Knol et al., 2011). The mag-
nitudes of the trend in $\Phi_{CO_2}$ over the 1980–2005 period for CanESM4.2 simulations with $\gamma_d = 0.4$ and $0.55$ ($0.328 \pm 0.038$ and $0.314 \pm 0.034 \, \text{ppm} \, (10 \, \text{yr})^{-1}$, respectively) are, however, more than twice, and statistically different from the observation-based estimate ($0.142 \pm 0.08 \, \text{ppm} \, (10 \, \text{yr})^{-1}$).

Overall, the CanESM2 simulation with $\gamma_d = 0.25$ yields the amplitude of the globally-average annual $CO_2$ cycle and its rate of increase over the 1980–2005 period that compares best with observation-based estimates.

## 4.3 Historical simulations without LUC

Figures 10 and 11 show results from CanESM4.2 emissions-driven simulations for three different strengths of the $CO_2$ fertilization effect that do not implement anthro-pogenic LUC over the historical period and compare them to their corresponding sim-ulations with LUC.

Discussion Paper | Discussion Paper | Discussion Paper | Discussion Paper | Discussion Paper |

**GMDD**

doi:10.5194/gmd-2015-252

**On constraining the strength of the terrestrial CO₂ fertilization effect**

V. K. Arora and J. F. Scinocca

Discussion Paper | Discussion Paper | Discussion Paper | Discussion Paper | Discussion Paper |

**[GMDD](doi:10.5194/gmd-2015-252)**

doi:10.5194/gmd-2015-252

**On constraining the strength of the terrestrial CO₂ fertilization effect**

V. K. Arora and
J. F. Scinocca

Figure 10a compares the simulated [CO₂]; as expected in the absence of anthropogenic LUC the simulated [CO₂] is lower since LUC emissions do not contribute to increase in [CO₂]. The difference in [CO₂] at the end of the simulation, in year 2005, between simulations with and without LUC is 29.0, 23.6 and 19.0 ppm for $\gamma_d = 0.25$,

0.40 and 0.55. The simulations with the lowest strength of the CO₂ fertilization effect ($\gamma_d = 0.25$) yield the largest difference because these simulations also have the largest [CO₂] amongst their set of simulations with and without LUC. The CO₂ fertilization of the terrestrial biosphere implies that the effect of deforestation will be higher, because of reduced carbon uptake by deforested vegetation, if background [CO₂] is higher.

Figure 10b compares the simulated NPP from CanESM2 simulations with and without LUC. The increase in simulated NPP, regardless of the strength of the CO₂ fertilization effect, is lower over the historical period in simulations without LUC for two apparent reasons. First, the rate of increase of [CO₂] is itself lower and second, in the absence of LUC, there is no contribution from increasing crop area to NPP. Overall,

the increase in NPP over the 1850–2005 period in simulations with LUC is a little more than twice that in simulations without LUC. Figure 10c and d compare the changes in global vegetation biomass and soil carbon mass, over the historical period, from simulations with and without LUC. As expected, in the absence of LUC, global vegetation biomass and soil carbon mass more or less show a continuous increase, associated

with the increase in NPP which itself is due to the increase in [CO₂]. Consequently, in Fig. 11a, the cumulative atmosphere–land CO₂ flux $\tilde{F}_L$ in simulations without LUC also shows a more or less continuous increase over the historical period.

Finally, Fig. 11b shows the diagnosed cumulative LUC emissions $\tilde{E}_L$ calculated as the difference between cumulative $\tilde{F}_L$, following Eq. (10), from simulations with and

without LUC. The calculated diagnosed $\tilde{E}_L$ in this manner are equal to 95, 81 and 67 Pg C, over the 1850–2005 period, for $\gamma_d = 0.25$, 0.40 and 0.55. The calculated diagnosed $\tilde{E}_L$ are highest for $\gamma_d = 0.25$ associated with the highest background simulated [CO₂] in these simulations, as mentioned earlier. For comparison, LUC emissions estimated by Houghton (2008) for the period 1850–2005, based on a book-keeping approach, are

156 Pg C but these estimates are generally believed to be ±50 % uncertain (see Fig. 1 of Ramankutty et al., 2007).

## 5 Discussion and conclusions

This study evaluates the ability of four observation-based determinants of the global carbon cycle and the historical carbon budget to constrain the parameterization of photosynthesis down-regulation, which directly determines the strength of the $CO_2$ fertilization effect, over the historical period 1850–2005. The key parameter that controls photosynthesis down-regulation in CTEM, $\gamma_d$, was varied in the latest version of CCCma's earth system model CanESM4.2. Comparing simulated and observation-based estimates of (1) globally-averaged atmospheric $CO_2$ concentration, (2) cumulative atmosphere–land $CO_2$ flux, and (3) atmosphere–land $CO_2$ flux for the decades of 1960s, 1970s, 1980s, 1990s and 2000s, it is found that the CanESM4.2 version with $\gamma_d = 0.40$ yields the best comparison.

CanESM4.2 simulates globally-averaged near-surface $[CO_2]$ of 400, 381 and 368 ppm for $\gamma_d = 0.25$, 0.40 and 0.55, respectively, compared to the observation-based estimate of 379 ppm for year 2005. The cumulative atmosphere–land $CO_2$ flux of 18 Pg C for the period 1850–2005 for $\gamma_d = 0.40$ lies within the range of the observation-based estimate of $-11 \pm 47$ Pg C in Fig. 4b, and so do the average atmosphere–land $CO_2$ flux for the decades of 1960s through to 2000s in Fig. 4a when compared to observation-based estimates from Le Quéré et al. (2015). $\gamma_d = 0.25$ and 0.55 yield average atmosphere–land $CO_2$ flux for the decades of 1960s through to 2000s that are lower and higher, respectively, than the observation-based estimates from Le Quéré et al. (2015). The only determinant against which $\gamma_d = 0.40$ does not yield the best comparison with observation-based estimates is the amplitude of the globally-averaged annual $CO_2$ cycle and its increase over the 1980 to 2005 period. For this determinant, $\gamma_d = 0.25$ seems to yield the best comparison (Fig. 9).

Discussion Paper | Discussion Paper | Discussion Paper | Discussion Paper | Discussion Paper |

**GMDD**

doi:10.5194/gmd-2015-252

**On constraining the strength of the terrestrial CO2 fertilization effect**

V. K. Arora and
J. F. Scinocca

The caveat with the analyses presented here, or for any model for that matter, is that the strength of the terrestrial $CO_2$ fertilization effect is dependent on the processes included in the model and the parameter values associated with them. The primary example of this is the adjustment to the humification factor in CTEM4.2, which leads to reduction in the global soil carbon amount as anthropogenic LUC becomes significant towards the mid-20th Century. This response of soil carbon was not present in the model's configuration of CTEM and historical simulations made with CanESM2. The representation of soil carbon loss, in response to anthropogenic LUC in CanESM4.2, implies that a stronger $CO_2$ fertilization effect (or weaker photosynthesis down-regulation) should be required to reproduce realistic atmosphere–land $CO_2$ flux over the historical period and this was found to be the case in Fig. 4a. Despite this dependence on processes included in the model, the response of the land carbon cycle, over the historical period, to the two primary forcings of increased [$CO_2$] and anthropogenic land use change must be sufficiently realistic in the model to satisfy all the four determinants of the global carbon cycle and the historical global carbon budget.

The simulated loss in soil carbon in response to anthropogenic LUC over the historical period may also be assessed against observation-based estimates from Wei et al. (2014). Using data from 453 sites that were converted from forest to agricultural land, Wei et al. (2014) find that the soil organic carbon stocks decreased by an average of $43.1 \pm 1.1$ % for all sites. Based on the HYDE v3.1 data set from which the changes in crop area are derived (Hurtt et al., 2011), LUC as implemented in CanESM4.2 yields an increase in crop area from about $5$ million $km^2$ in 1850 to about $15$ million $km^2$ in 2005. Assuming an initial soil carbon amount of $10 \, kg \, C \, m^{-2}$ (see Fig. 2c of Melton and Arora, 2014) and an average 40 % decrease in soil carbon amount, based on Wei et al. (2014), implies that the increase in crop area of about $10$ million $km^2$ over the historical period has likely yielded a global soil organic carbon loss of 40 Pg C. The loss in soil carbon in Fig. 5a is simulated to 18 Pg C for CanESM4.2 simulation with $\gamma_d = 0.40$, the simulation that yield best comparison with observation-based determinants of the global carbon cycle and the historical carbon budget. This loss of 18 Pg C is expected

**GMDD**

doi:10.5194/gmd-2015-252

**On constraining the strength of the terrestrial CO₂ fertilization effect**

V. K. Arora and
J. F. Scinocca

to be less than the 40 Pg C because the model estimates also include an increase associated with the increase in NPP due to the $CO_2$ fertilization effect from non-crop areas. The effect of LUC on global soil carbon loss may also by estimated by differencing global soil carbon amounts from simulations with and without LUC from Fig. 10d at the end of the simulation in year 2005. For CanESM2 simulation with $\gamma_d = 0.40$, this amounts to around 50 Pg C. Both these estimates of soil carbon loss are broadly consistent with the back-of-the-envelope calculation of 40 Pg C soil carbon loss, based on Wei et al. (2014) estimates, indicating that the soil carbon loss simulated in response to anthropogenic LUC over the historical period is not grossly over or underestimated.

The CanESM4.2 simulation with $\gamma_d = 0.40$, however, fails to satisfy the rate of increase of the amplitude of the globally-averaged annual $CO_2$ cycle over the 1980–2005 period implying that there are still limitations in the model structure and/or parameter values. Of course, the fact that the amplitude of the globally-averaged annual $CO_2$ cycle is also affected by the atmosphere–ocean $CO_2$ fluxes makes it more difficult to attribute the changes in the amplitude of the globally-averaged annual $CO_2$ cycle solely to atmosphere–land $CO_2$ fluxes. Additionally, the increase in crop area as well as crop yield per unit area over the historical period have been suggested by Zeng et al. (2014) to contribute towards the observed increase in the amplitude of annual $CO_2$ cycle. Based on their sensitivity tests, Zeng et al. (2014) attribute 45, 29 and 26 % of the observed increase in the seasonal-cycle amplitude of the $CO_2$ cycle to LUC, climate variability and change (including factors such as the lengthening of the growing season) and increased productivity due to $CO_2$ fertilization, respectively. Comparison of the rate of increase of NPP in CanESM4.2 experiments with and without LUC (Fig. 10b), as a measure of increase in the strength of the $CO_2$ fertilization effect, suggests that the contribution of anthropogenic LUC to the increase in the seasonal-cycle amplitude is 52 %, which is broadly consistent with the 45 % value obtained by Zeng et al. (2014).

While CanESM4.2 simulation with $\gamma_d = 0.40$ is able to simulate a realistic rate of increase of [$CO_2$] over the period 1960 to 2005, the modelled atmosphere–ocean $CO_2$ fluxes for this and the CanESM2 version are lower than observational estimates of this

Discussion Paper | Discussion Paper | Discussion Paper | Discussion Paper |

**GMDD**

doi:10.5194/gmd-2015-252

**On constraining the strength of the terrestrial CO₂ fertilization effect**

V. K. Arora and J. F. Scinocca

quantity (Fig. 8). This implies that if the modelled atmosphere–ocean $CO_2$ flux were to increase and become more consistent with observation-based estimates then the modelled atmosphere–land $CO_2$ flux must decrease to still be able to yield sufficiently realistic rate of increase of $[CO_2]$. This implies that the strength of the terrestrial $CO_2$

fertilization effect should likely be somewhat lower than what is obtained by $\gamma_d = 0.40$ or the simulated atmosphere–land $CO_2$ flux is higher because of some other reason, most likely lower LUC emissions. Indeed, the required decrease in modelled atmosphere–land $CO_2$ flux is consistent with the fact that the modelled LUC emissions for $\gamma_d = 0.40$ (81 Pg C) are about half the estimate from Houghton (2008) (156 Pg C) with the caveat,

of course, that Houghton's estimates themselves have an uncertainty of roughly ±50 %. The LUC module of CTEM currently only accounts for changes in crop area and does not take into account changes associated with pasture area given their ambiguous definition (pasture may or may not be grasslands). The model also does not take into account wood harvesting which amongst other uses is also used as a biofuel. Treat-

ment of these additional processes will increase modelled LUC emissions.

Although the CanESM4.2 simulation with $\gamma_d = 0.40$ satisfies three out of four constraints placed by the chosen determinants of the global carbon cycle and the historical carbon budget, and also simulates reasonable soil carbon loss in response to anthropogenic LUC, the model now yields the highest land carbon uptake, in the 1ptCO2

experiment, amongst the CMIP5 models that were compared by Arora et al. (2013) as seen in Fig. 2. It is quite possible that the chosen determinants of the global carbon cycle and the historical carbon budget are not able to constrain the model sufficiently, given the especially large uncertainty associated with LUC emissions. Nevertheless, these observation-based constraints of the carbon cycle and historical carbon budget

are essentially the only means to evaluate carbon cycle aspects of the ESMs at the global scale including the strength of the terrestrial $CO_2$ fertilization effect. In the near future, availability of model output from the sixth phase of CMIP (CMIP6) will allow a comparison of the simulated aspects of the global carbon cycle and the historical carbon budget from ESMs to observations-based estimates for the 1850–2014 period.

**GMDD**

doi:10.5194/gmd-2015-252

**On constraining the strength of the terrestrial CO2 fertilization effect**

V. K. Arora and
J. F. Scinocca

These data will allow a comparison of the rate of increase of the amplitude of globally-averaged surface [$CO_2$] in models with observation-based estimates over a longer period. This should help better constrain the strength of the terrestrial $CO_2$ fertilization effect, as it is represented in models, in a somewhat more robust manner.

# Source code and data availability

Source code for the complete CanESM4.2 model is an extremely complex set of FORTRAN subroutines, with C preprocessor (CPP) directives, that reside in CCCma libraries. Unix shell scripts process the model code for compilation based on CPP directives and several other switches (e.g. those related to free-$CO_2$, specified-$CO_2$, and relaxed-$CO_2$ settings). As such, it is extremely difficult to make the full model code available. However, selected model subroutines related to specific physical and biogeochemical processes can be made available by either author (vivek.arora@canada.ca, john.scinocca@canada.ca) upon agreeing to Environment and Climate Change Canada's software licensing agreement available at http://collaboration.cmc.ec.gc.ca/science/rpn.comm/license.html. Data used to produce plots and figures can be obtained from the first author (vivek.arora@canada.ca).

## Copyright statement

Discussion Paper | Discussion Paper | Discussion Paper | Discussion Paper | Discussion Paper |

**GMDD**

doi:10.5194/gmd-2015-252

**On constraining the strength of the terrestrial CO₂ fertilization effect**

V. K. Arora and J. F. Scinocca

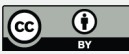

*Acknowledgements.* Authors would like to thank Joe Melton and Neil Swart for providing comments on an earlier version of this paper. Support and help from other members of the CanESM development team is also acknowledged including, in alphabetical order, Slava Kharin, Mike Lazare, and Larry Solheim.

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

Discussion Paper | Discussion Paper | Discussion Paper | Discussion Paper | Discussion Paper

**Table 1.** Summary of simulations performed for this study and the forcings used.

| Simulation | 1pctCO2 | esmhistorical | esmhistorical_noluc |
|---|---|---|---|
| Simulation details | 1 % year$^{-1}$ increasing CO$_2$ simulation | 1850–2005 historical simulation based on CMIP5 protocol. | 1850–2005 historical simulation based on CMIP5 protocol, but with no anthropogenic land use change. |
| Length | 140 years | 156 years | |
| CO$_2$ forcing | 285 ppm at the start of the simulation and 1140 ppm after 140 years. | Historical CO$_2$ forcing | |
| Land cover forcing | Land cover corresponds to its 1850 state. | Land cover evolution is based on increase in crop area over the historical period. | Land cover corresponds to its 1850 state. |
| Non-CO$_2$ greenhouse gases forcing | Concentration of non-CO$_2$ GHGs is specified at their 1850 levels. | Concentration of non-CO$_2$ GHGs is specified and evolves over the historical period based on the CMIP5 protocol. | |
| Aerosols forcing | Emissions of aerosols and their precursors are specified at their 1850 levels. | Emissions of aerosols and their precursors are specified and evolve over the historical period based on the CMIP5 protocol. | |

Discussion Paper | Discussion Paper | Discussion Paper | Discussion Paper | Discussion Paper |

**GMDD**

doi:10.5194/gmd-2015-252

**On constraining the strength of the terrestrial CO$_2$ fertilization effect**

V. K. Arora and J. F. Scinocca

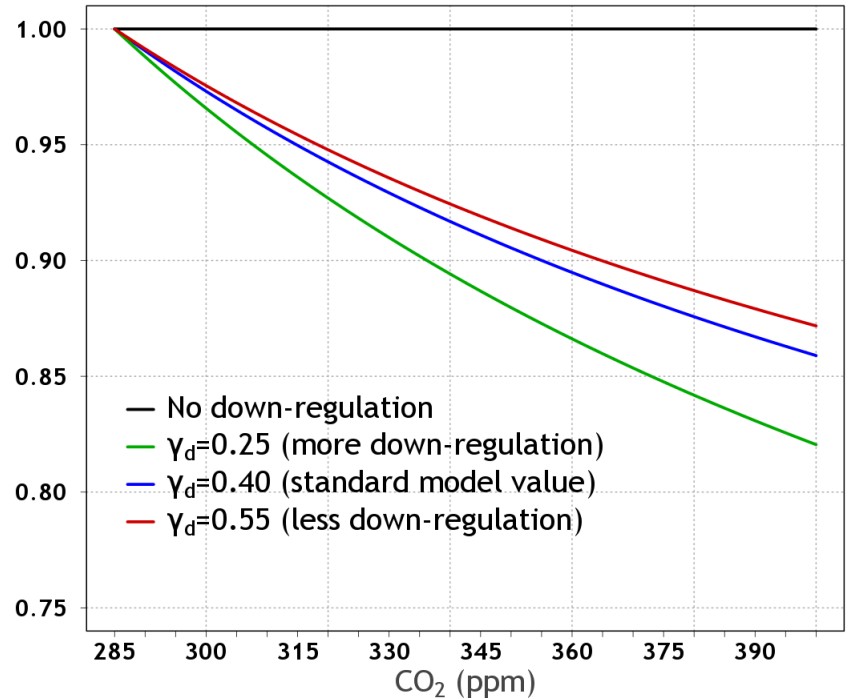

Down-regulation factor as a function of $CO_2$ concentration

Legend:
— No down-regulation
— $\gamma_d$=0.25 (more down-regulation)
— $\gamma_d$=0.40 (standard model value)
— $\gamma_d$=0.55 (less down-regulation)

$CO_2$ (ppm)

**Figure 1.** The behaviour of terrestrial photosynthesis down-regulation scalar $\xi(C)$ (Eq. 11) for $\gamma_p$ = 0.95 and values of $\gamma_d$ equal to 0.25, 0.4 and 0.55 that are used in CanESM4.2 simulations.

Discussion Paper | Discussion Paper | Discussion Paper | Discussion Paper |

GMDD

doi:10.5194/gmd-2015-252

On constraining the strength of the terrestrial $CO_2$ fertilization effect

V. K. Arora and J. F. Scinocca

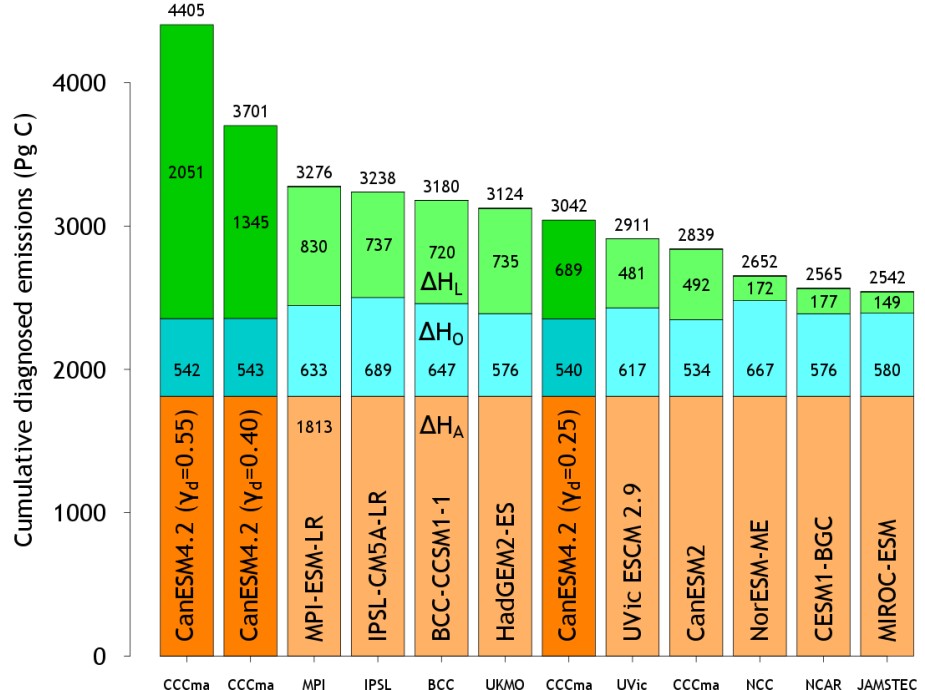

**Figure 2.** Components of the carbon budget Eq. (8) that make up cumulative diagnosed emissions based on results from the fully-coupled 1pctCO2 experiment. Results shown are from eight CMIP5 models that participated in the Arora et al. (2013) study and from three CanESM4.2 simulations (shown in darker colours) for three different strengths of the terrestrial $CO_2$ fertilization effect.

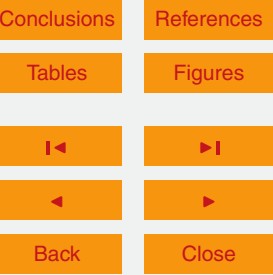

# GMDD

doi:10.5194/gmd-2015-252

## On constraining the strength of the terrestrial CO$_2$ fertilization effect

V. K. Arora and
J. F. Scinocca

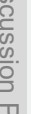

Discussion Paper | Discussion Paper | Discussion Paper | Discussion Paper | Discussion Paper |

# GMDD

doi:10.5194/gmd-2015-252

**On constraining the strength of the terrestrial CO₂ fertilization effect**

V. K. Arora and
J. F. Scinocca

Model minus Xie and Arkin precipitation
averaged over the 1979-1998 period (mm/day)

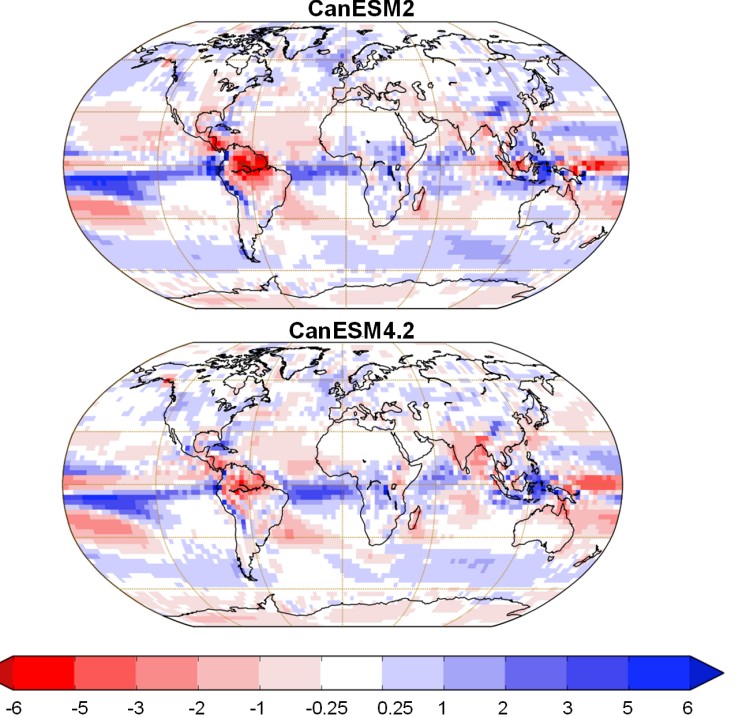

**Figure 3.** CanESM2 **(a)** and CanESM4.2 (**b**, $\gamma_d$ = 0.40) precipitation anomalies compared to the observation-based estimates from CPC Merged Analysis of Precipitation (CMAP) based on Xie and Arkin (1997) averaged over the 1979–1998 period.

Discussion Paper | Discussion Paper | Discussion Paper | Discussion Paper |

**GMDD**

doi:10.5194/gmd-2015-252

**On constraining the strength of the terrestrial CO$_2$ fertilization effect**

V. K. Arora and
J. F. Scinocca

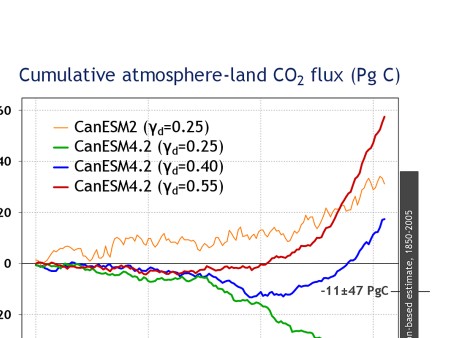

**Figure 4.** Atmosphere–land CO$_2$ flux ($F_L$) **(a)** and its cumulative values $\tilde{F}_L$ **(b)** from CanESM2 and the three CanESM4.2 historical 1850–2005 simulations for different strengths of the terrestrial CO$_2$ fertilization effect. In **(a)** the observation-based estimates of $F_L$ and their uncertainty, show via boxes, for the decades of 1960, 1970, 1980, 1990 and 2000 are reproduced from Le Quéré et al. (2015). The bold lines in **(a)** are the 10-year moving averages of the annual $F_L$ values which are shown in light colours. The results from CanESM2 and CanESM4.2 are the average of the two ensemble members.

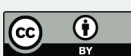

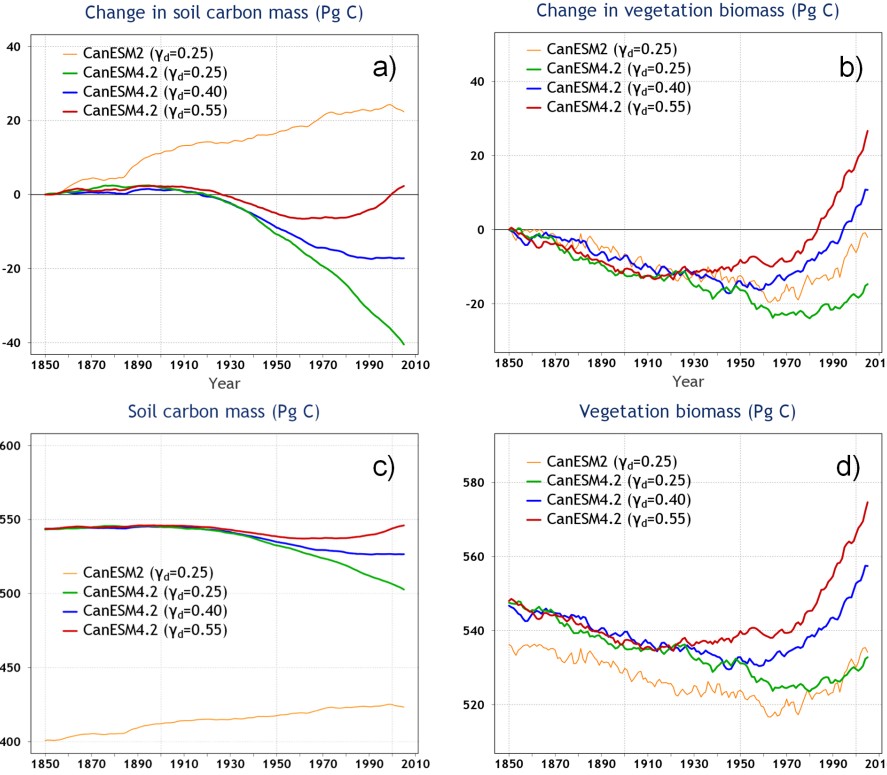

**Figure 5.** Change in and absolute values of global soil carbon and vegetation biomass amounts from CanESM2 and the three CanESM4.2 historical 1850–2005 simulations with different strengths of the terrestrial CO$_2$ fertilization effect. The results shown in all panels are the average of the two ensemble members.

Discussion Paper | Discussion Paper | Discussion Paper | Discussion Paper | Discussion Paper |

**GMDD**

doi:10.5194/gmd-2015-252

**On constraining the strength of the terrestrial CO$_2$ fertilization effect**

V. K. Arora and
J. F. Scinocca

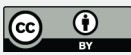

Discussion Paper | Discussion Paper | Discussion Paper | Discussion Paper | Discussion Paper |

# GMDD

doi:10.5194/gmd-2015-252

## On constraining the strength of the terrestrial CO₂ fertilization effect

V. K. Arora and
J. F. Scinocca

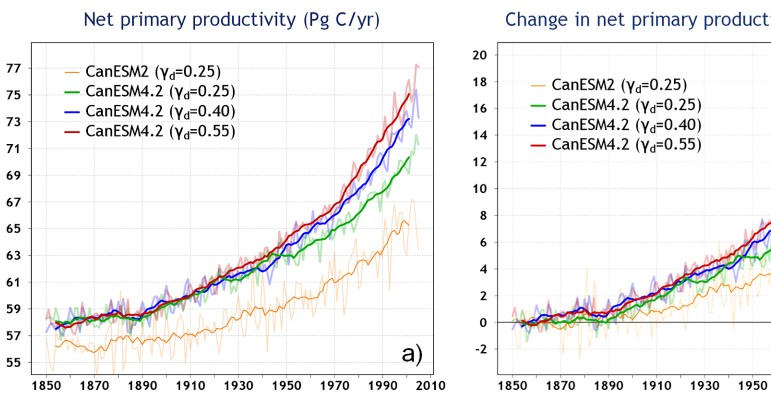

**Figure 6.** Absolute values of **(a)**, and change in **(b)**, net primary productivity (NPP) from CanESM2 and the three CanESM4.2 historical 1850–2005 simulations with different strengths of the terrestrial CO₂ fertilization effect. The thin lines show the ensemble-mean based on results from the two ensemble members and the bold lines are their 10-year moving averages.

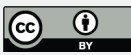

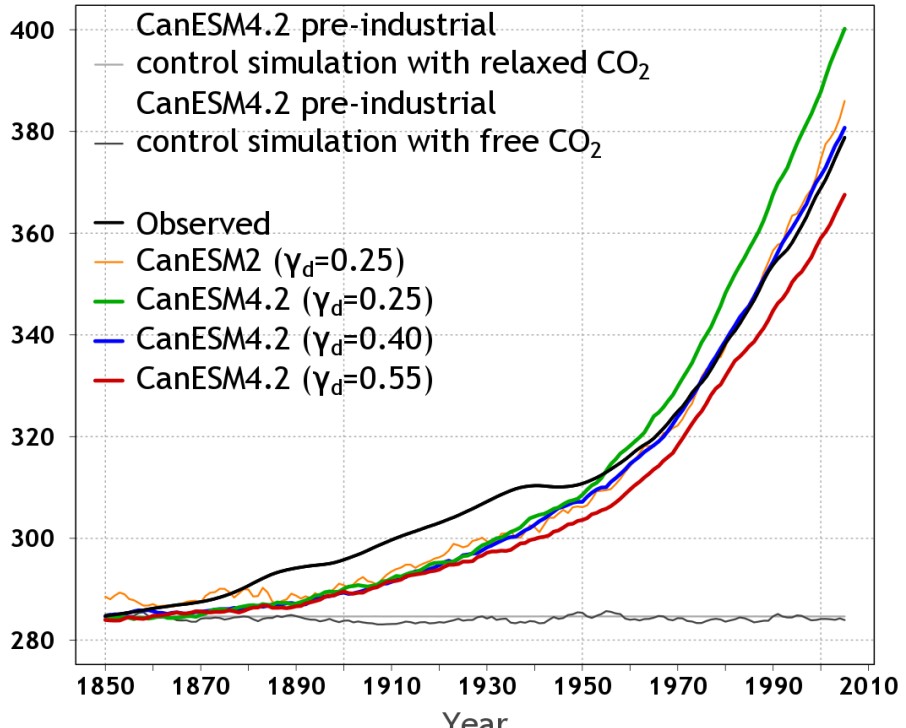

**Figure 7.** Simulated globally-averaged surface atmospheric $CO_2$ concentration from CanESM2 and the three CanESM4.2 historical 1850–2005 simulations with different strengths of the terrestrial $CO_2$ fertilization effect. The observation-based concentration is shown in black. Also shown is the $CO_2$ concentration of 284.6 ppm used in CanESM4.2's pre-industrial simulation with "relaxed" $CO_2$ and the simulated concentration from the pre-industrial CanESM4.2 simulation with interactively determined $CO_2$.

Discussion Paper | Discussion Paper | Discussion Paper | Discussion Paper | Discussion Paper |

**GMDD**

doi:10.5194/gmd-2015-252

**On constraining the strength of the terrestrial CO₂ fertilization effect**

V. K. Arora and
J. F. Scinocca

Discussion Paper | Discussion Paper | Discussion Paper | Discussion Paper | Discussion Paper |

**GMDD**

doi:10.5194/gmd-2015-252

**On constraining the strength of the terrestrial CO$_2$ fertilization effect**

V. K. Arora and
J. F. Scinocca

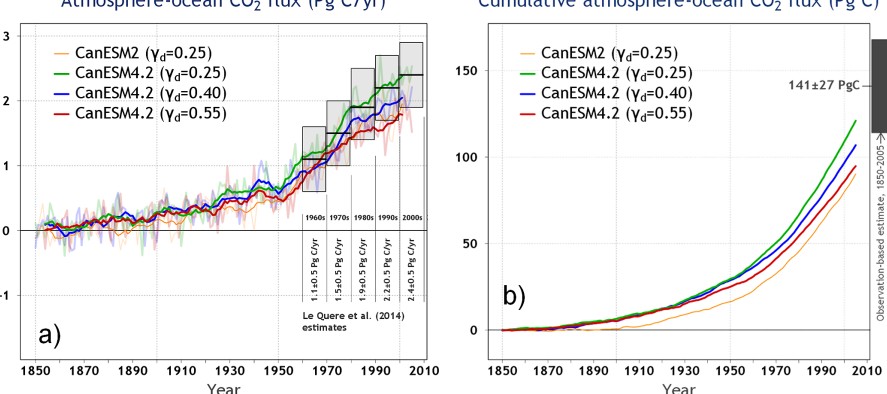

**Figure 8.** Atmosphere–ocean CO$_2$ flux ($F_O$) **(a)** and its cumulative values $\tilde{F}_O$ **(b)** from CanESM2 and the three CanESM4.2 historical 1850–2005 simulations three different strengths of the terrestrial CO$_2$ fertilization effect. In **(a)** the observation-based estimates of $F_O$ and their uncertainty, show via boxes, for the decades of 1960, 1970, 1980, 1990 and 2000 are reproduced from Le Quéré et al. (2015). The bold lines in **(a)** are the 10-year moving averages of the annual $F_O$ values which are shown in light colours. The results from CanESM2 and CanESM4.2 are the average of the two ensemble members.

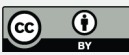

Discussion Paper | Discussion Paper | Discussion Paper | Discussion Paper | Discussion Paper |

**GMDD**

doi:10.5194/gmd-2015-252

**On constraining the strength of the terrestrial CO$_2$ fertilization effect**

V. K. Arora and J. F. Scinocca

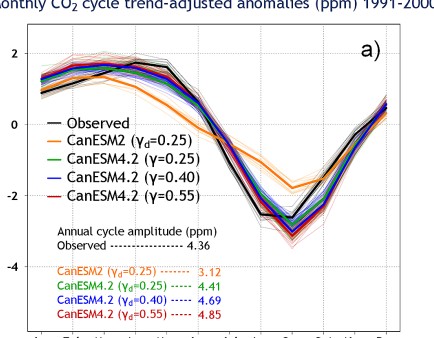

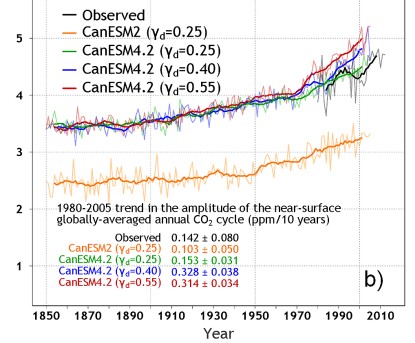

**Figure 9.** The annual cycle of trend-adjusted globally-averaged near-surface monthly [CO$_2$] anomalies from CanESM2, the versions of CanESM4.2 for three different strengths of the CO$_2$ fertilization effect and observation-based estimates for the 1991–2000 period **(a)**. **(b)** shows the time series of the amplitude of the annual cycle of the trend adjusted globally-averaged near-surface monthly [CO$_2$] anomalies for corresponding model and observation-based estimates. The bold lines are 10-year moving averages and the thin lines for model results are the average of results from two ensemble members.

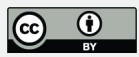

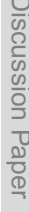



**Figure 10.** Comparison of CanESM4.2 simulations with and without implementation of anthropogenic land use change over the historical period for three different strengths of the terrestrial $CO_2$ fertilization effect: **(a)** globally-averaged annual surface atmospheric $CO_2$ concentration, **(b)** net primary productivity, **(c)** global vegetation biomass, and **(d)** global soil carbon mass. All lines are the average of results from two ensemble members. Additionally, in **(b)** the bold lines are the 10-year moving averages.

# GMDD

doi:10.5194/gmd-2015-252

## On constraining the strength of the terrestrial CO₂ fertilization effect

V. K. Arora and
J. F. Scinocca

Discussion Paper | Discussion Paper | Discussion Paper | Discussion Paper

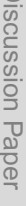

**GMDD**

doi:10.5194/gmd-2015-252

**On constraining the strength of the terrestrial CO$_2$ fertilization effect**

V. K. Arora and
J. F. Scinocca

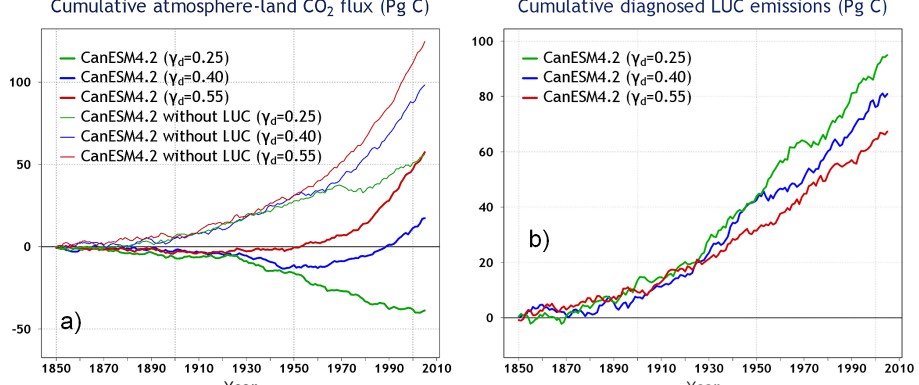

**Figure 11.** Comparison of simulated cumulative atmosphere–land CO$_2$ flux from CanESM4.2 simulations with and without implementation of anthropogenic land use change over the historical period for three different strengths of the terrestrial CO$_2$ fertilization **(a)**. **(b)** shows the cumulative diagnosed LUC emissions calculated using Eq. (10) as the difference between cumulative atmosphere–land CO$_2$ flux from simulations with and without LUC shown in **(a)**. All lines are the average of results from two ensemble members.

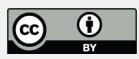