# Peer review of "Constraining the strength of the terrestrial CO2 fertilization effect in the Canadian Earth System Model version 4.2 (CanESM4.2)"

_Geoscientific Model Development, 2015_

## Referee Comment (RC1) · Anonymous Referee #1 · 29 Feb 2016

Comments on "On constraining the strength of the terrestrial CO2 fertilization effect in an Earth system model" submitted by V.K.Arora and J.F.Scinocca to Geoscientific Model Development

General comments

In this manuscript, the authors attempted to constrain a parameter of the Canadian Earth System Model version 4.2 in terms of atmospheric CO2 fertilization effect, which is one of the most uncertain process in the future climate–carbon cycle feedback. By conducting a series of simulations using different parameter values (gamma-d = 0.25, 0.4, 0.55), they chose the most plausible parameter value that allows most realistic simulations of atmospheric CO2 growth and its seasonal amplitude. Apparently, this is

an up-to-date and meaningful work to improve the reliability of Earth System Models. The new experiment, "relaxed-$CO_2$", is especially interesting for me. The manuscript was clearly written and I found no logical fault. Nevertheless, I have a few moderate caveats on this study. First, the $CO_2$ fertilization parameter (gamma-d) represents photosynthetic down-regulation (not the fertilization effect itself) in an empirical manner. So, the selected parameter value (i.e., 0.4) seems to be specific to the CanESM4.2. Second, this study compared only three parameter values, and so the selected one (0.4) may not be exactly the best one. Third, recently, Schimel et al. (2015) published a very relevant paper on constraining the $CO_2$ fertilization effect, but this was not referred in the manuscript. In conclusion, the manuscript is well prepared and may be accepted for publication after moderate revision. Specific comments are given below.

Specific comments

Page 4 Line 21–26: Several studies used FACE data for benchmarking of terrestrial vegetation models (Piao et al., 2013; Zaehle et al., 2014).

Page 12 Line 24: How the default parameter of CanESM2 (gamma-d = 0.25) was determined?

Page 19 Line 25: Remove the space between "under" and "predict".

References

Piao, S., Sitch, S., Ciais, P., Friedlingstein, P., Peylin, P., Wang, X., Ahlström, A., Anav, A., Canadell, J. G., Cong, N., Huntingford, C., Jung, M., Levis, S., Levy, P. E., Li, J., Lin, X., Lomas, M. R., Lu, M., Luo, Y., Ma, Y., Myneni, R. B., Poulter, B., Sun, Z., Wang, T., Viovy, N., Zaehle, S., and Zeng, N.: Evaluation of terrestrial carbon cycle models for their response to climate variability and to $CO_2$ trends, Global Change Biol., 19, 2117–2132, doi:10.1111/gcb.12187, 2013.

Schimel, D., Stephens, B. B., and Fisher, J. B.: Effect of increasing $CO_2$ on the terrestrial carbon cycle, Proceedings of the National Academy of Science U.S.A., 112,

436–441, doi:10.1073/pnas.1407302112, 2015.

Zaehle, S., Medlyn, B. E., De Kauwe, M. G., Walker, A. P., Dietze, M. C., Hickler, T., Luo, Y., Wang, Y.-P., El-Masri, B., Thornton, P., Jain, A., Wang, S., Warlind, D., Weng, E., Parton, W., Iversen, C. M., Gallet-Budynek, A., McCarthy, H., Finzi, A., Hanson, P. J., Prentice, I. C., Oren, R., and Norby, R.: Evaluation of 11 terrestrial carbon–nitrogen cycle models against observations from two temperate Free-Air CO2 Enrichment studies, New Phytologist, 202, 803–822, doi:10.1111/nph.12697, 2014.

---

## Referee Comment (RC2) · Anonymous Referee #2 · 5 Mar 2016

[General comments]

Authors present in this paper the structure of the new Earth system model developed in CCCma, and then they attempt to evaluate the model's performance to reproduce the global carbon budget and atmospheric CO2 concentration during 1850-2005 periods, with simulation ensembles and different parameters/configurations. In their evaluation, they focus on particularly the land ecosystem process so called "CO2 fertilization effect", which is strongly associated with the most uncertain feedback process within the global carbon cycle. It is noteworthy that the authors consider four types of observation constraints in their model evaluation, which makes their conclusions more robust. Overall, this paper is clearly written and well structured, and will contribute to the journal. Detailed comments are listed below, and I believe most of them will not require much effort to improve.

[Detailed comments]

p4, L7- "the uncertainty in the carbon-concentration feedback over land had somewhat reduced since the first coupled carbon cycle climate model intercomparison project (C4MIP)" I'm afraid this sentence might mislead readers. Since the 1st and 2nd MIP used different scenarios (SRES-A2 / 1pctCO2) and configurations (emission/concentration-driven) to evaluate carbon cycle feedbacks, we cannot directly compare the feedback strength between the two MIPs.

P10, L23-

It will be helpful for readers to briefly mention the decay-timescale of the pools for "short" and "long" (: from Arora and Boer 2011, it seems the two product pools are equivalent to litter/soil). This information will be helpful to understand the reduction of soil carbon mass in LUC simulation and the delayed response of soil carbon pools (Fig. 5c).

P23 L19; p24 L10; p27 L5

Should these "CanESM2" be replaced by "CanESM4.2" ?

P24 L28- p25 L2

In my understanding, your choice of "emission-driven" configuration might be one of the reasons to underestimate the LUC emission (E~L): since LUC emission is omitted in the "without LUC experiments", the CO2 concentration stays lower level and the CO2 fertilization effect becomes weaker. As a result, the cumulative land carbon uptake in the "without LUC" experiment (FL') is more or less underestimated, which yields lower E~L (=FL' - FL). I recommend the authors to mention this.

Discussion section

As commented above, simulations without LUC inevitably lead to lower CO2 concentration and weaker CO2 fertilization effect. I think this can be a "noise" when evaluating LUC emission/impacts. Specifically, in Fig.4(b), NPP in "without LUC" simulation are generally lower than "with LUC", but it is difficult to identify the reason of the difference, because the NPP difference can be affected by CO2 fertilization, increased GPP by crops, and vegetation regrowth. I hope the authors to make a few discussions about the configuration settings for evaluating LUC impacts. I believe such information will be helpful when making simulation designs in the coming CMIP.

In Fig.2, Gamma_d=0.25 simulations display moderate land carbon sink among CMIP5-ESMs. I think this result is reasonable because most CMIP5-ESMs may not consider down-regulation mechanism; Fig.9 also supports the choice of the parameter value. However, the historical simulations with gamma_d=0.25 did not do a good job for reproducing land carbon uptake (Fig. 4). Although you discussed on this in the text, I suppose we have two more things to discuss. The first is the additional carbon uptake by vegetation regrowth. Although the regrowth mechanisms in the model are presented on p10-11, I'm not sure if the modeling was appropriate or not. If we can expect more carbon gain by vegetation regrowth, simulations with gamma_d=0.25 may work better. The second is the parameter value of humification factor. If you choose more moderate value for the humification factor (or modify the fractions of deforested/removed biomass that goes into fast/slow pools), soil carbon mass displayed in Fig. 5c will push up toward positive, and this treatment will also make the simulation with gamma_d=0.25 more realistic. . . I hope to see some discussions on these two points.

About Title:

I'm thinking the key feature of this paper is constraining the historical carbon budget of the model from different angles. Of course, it is necessary for your model to choose an appropriate value for the down-regulation, but its parameterization looks somewhat specific to your model. My suggestion is to change the title to reflect "CO2 fertilization", "LUC", and "historical carbon budget": I believe these are the main issues in the

background and will have more meaningful messages for readers.

---

## Referee Comment (RC3) · Anonymous Referee #3 · 8 Mar 2016

Comments:

Overall comment: This is an interesting study and asks an important question: how can we constrain an emergent propoerty such as the global responsiveness to elevated CO2 based on global transient observational records? The authors are careful to emphasize the contingent nature of their answers, and emphasize that such answers cannot be unambiguously identified by this approach due to the presence of large amounts of model and forcing uncertainty that determine the response. I would personally go further and ask whether it makes sense at all to try to "tune" emergent model properties to match transient data in such an explicit manner. The more widely-followed approach is to test model components at scales where process-level

understanding can be gained, in hopes of removing some of the dependence on over-all model behavior that may influence results, for example by comparing at FACE sites (e.g. the various FACE-MIP papers), or by systematically benchmarking multiple aspects of the model in order to better understand the structural control on emergent behaviors. So while I do see this paper as a valuable contribution, I also feel that, in the end, the answer to the problem posed in the title is that it very much depends on what is in the ESM itself, and so without understanding how accurate the model is across a wide range of predictions, it is impossible to know whether the specific answer inferred by the comparison is informative of the real world or not.

page 5, lines 7-14: I don't see how, from the perspective of the terrestrial biosphere, the information content of the first three of these tests are different. So if the focus is just on the land, why include the total CO2 growth rate at all since the answer is effected by the uncertainty in ocean and fossil fuel emissions?

page 14, discussion on "relaxed CO2" approach. This seems to be a side point that isn't fully explained here, and I suggest either going into a bit more detail of what you mean (with figures or schematics) or else delete. Is the point that when you run it with relaxed CO2, you are able to assess whether or ot the model is in equilibrium? Or is the point that the 3D structure and seasonal variation of the CO2 matters from a radiative perspective and therefore leads to a different baseline climate than in the specified CO2 case?

table 1: It might be useful to add a row here with the purpose of each scenario.

figure 1: Why is this functional form of the downregulaiton factor chosen? Assuming that the downregulation is meant to capture progressive nutrient limitation, it doesn't actually seem very progressive–the initial slope is quite high and then lessens at higher CO2, but wouldn't one expect a priori that nutient limitations ought to become stronger only at higher CO2 levels? Secondly, I can imagine that part of this phase space in this figure would be effectively excluded in that it would actually cause GPP to decrease

under elevated CO2, but it isn't apparent from the figure where that boundary would occur. have you performed sufficient sensitivity studies to identify where that transition is? Thirdly, the minimal downregulation case is quite close to the sgtandard model, why was that chosen?

page 28, last paragraph. Implicit in this argument seems to be the idea that the degree of historical growth and the response of the terrestrial biosphere over the historical period ought to be informative of an idealized 1%/yr forcing. But to the extent that downregulation is pregressively driven by nutrient limitations, it ought to be expressed differently based on the rate at which CO2 increases. So it may be just as informative to consider an extremely rapid CO2 increase even if not at global scale, as in a FACE experiment, as it is to consider the slower-than 1%/yr forcing that has been applied globally over the historical period.

---

## Author Comment (AC1) · 23 Mar 2016

We thank our reviewers for their constructive and detailed comments. Our responses to reviewers' comments are indicated in **bold font** and indented, while reviewers' comments are shown in a regular font. We will use our responses to reviewers' comments (shown below) to revise our manuscript.
* * *
Reviewer # 1

In this manuscript, the authors attempted to constrain a parameter of the Canadian Earth System Model version 4.2 in terms of atmospheric $CO_2$ fertilization effect, which is one of the most uncertain process in

the future climate–carbon cycle feedback. By conducting a series of simulations using different parameter values (gamma-d = 0.25, 0.4, 0.55), they chose the most plausible parameter value that allows most realistic simulations of atmospheric $CO_2$ growth and its seasonal amplitude. Apparently, this is an up-to-date and meaningful work to improve the reliability of Earth System Models. The new experiment, "relaxed-$CO_2$", is especially interesting for me. The manuscript was clearly written and I found no logical fault. Nevertheless, I have a few moderate caveats on this study.

First, the $CO_2$ fertilization parameter (gamma-d) represents photosynthetic down-regulation (not the fertilization effect itself) in an empirical manner. So, the selected parameter value (i.e., 0.4) seems to be specific to the CanESM4.2.

> **Thank you pointing this. Yes, indeed the $\gamma_d$ parameter indicates down-regulation and not the $CO_2$ fertilization which would be indicated by $(1 - \gamma_d)$. As we also mention later in response to reviewer # 3, while the parameter $\gamma_d$ is specific to our model, it is the rate of increase of NPP that is relevant to other modellers and the community at large.**

Second, this study compared only three parameter values, and so the selected one (0.4) may not be exactly the best one.

> **It wasn't our objective to run the model for tens of possible values of the $\gamma_d$ parameter. Rather, the objective of the manuscript is to illustrate how this parameter can be adjusted in the framework of our model to best reproduce aspects of the global carbon cycle and the historical carbon budget.**

Third, recently, Schimel et al. (2015) published a very relevant paper on constraining the $CO_2$ fertilization effect, but this was not referred in the manuscript.

> **The Schimel et al. (2015) paper focusses on the relative roles of tropical and extra-tropical terrestrial carbon sinks but does not explicitly attempts to constrain the strength of the $CO_2$ fertilization effect. Yet, it's a relevant paper and a suitable place to mention this is in the introductory section.**

In conclusion, the manuscript is well prepared and may be accepted for publication after moderate revision.

> **Thank you.**

Specific comments are given below.

Page 4 Line 21–26: Several studies used FACE data for benchmarking of terrestrial vegetation models (Piao et al., 2013; Zaehle et al., 2014).

**Reviewer # 3 mentions that the traditional and more widely followed approach of model evaluation and parameter calibration is where process level understanding can be gained. This is in contrast to our top-down kind of approach where we evaluate an emergent property at the global scale. We will use the above suggestion in that context and when addressing reviewer # 3's comments in revising our manuscript.**

Page 12 Line 24: How the default parameter of CanESM2 (gamma-d = 0.25) was determined?

**For CanESM2 we used only two determinants to determine the value of $\gamma_d$ parameter – globally averaged surface $CO_2$ and cumulative atmosphere-land $CO_2$ flux. CanESM2 wasn't as rigourously evaluated. We will clarify this.**

Page 19 Line 25: Remove the space between "under" and "predict".

**Thank you for noting this.**
* * *
Reviewer # 2

Authors present in this paper the structure of the new Earth system model developed in CCCma, and then they attempt to evaluate the model's performance to reproduce the global carbon budget and atmospheric $CO_2$ concentration during 1850-2005 periods, with simulation ensembles and different parameters/configurations. In their evaluation, they focus on particularly the land ecosystem process so called "$CO_2$ fertilization effect", which is strongly associated with the most uncertain feedback process within the global carbon cycle. It is noteworthy that the authors consider four types of observation constraints in their model evaluation, which makes their conclusions more robust. Overall, this paper is clearly written and well structured, and will contribute to the journal. Detailed comments are listed below, and I believe most of them will not require much effort to improve.

p4, L7- "the uncertainty in the carbon-concentration feedback over land had somewhat reduced since the first coupled carbon cycle climate model intercomparison project (C4MIP)" I'm afraid this sentence might mislead readers. Since the 1st and 2nd MIP used different scenarios (SRES-A2 / 1pctCO$_2$) and configurations (emission/concentration-driven) to evaluate carbon cycle feedbacks, we cannot directly compare the feedback strength between the two MIPs.

**Yes, it is true that the first C4MIP study was performed for the SRES A2 scenario while Arora et at. (2013) used results from the 1% per year increasing $CO_2$ simulation. This means that the average strengths of feedbacks cannot be compared across the two studies. However, our sentence attempts to compares the uncertainty in calculated values of the feedback**

**parameters as indicated by their standard deviation. We will clarify this.**

P10, L23- It will be helpful for readers to briefly mention the decay-timescale of the pools for "short" and "long" (: from Arora and Boer 2011, it seems the two product pools are equivalent to litter/soil). This information will be helpful to understand the reduction of soil carbon mass in LUC simulation and the delayed response of soil carbon pools (Fig. 5c).

**Yes, the short and long time scales for the land use change (LUC) products correspond to time scales of the litter and soil carbon pools.**

P23 L19; p24 L10; p27 L5 Should these "CanESM2" be replaced by "CanESM4.2" ?

**Thank you for noting these typos.**

P24 L28- p25 L2 In my understanding, your choice of "emission-driven" configuration might be one of the reasons to underestimate the LUC emission (EL): since LUC emission is omitted in the "without LUC experiments", the $CO_2$ concentration stays lower level and the $CO_2$ fertilization effect becomes weaker. As a result, the cumulative land carbon uptake in the "without LUC" experiment (FL') is more or less underestimated, which yields lower EL (=FL' - FL). I recommend the authors to mention this.

**When LUC emissions are determined by differencing atmosphere-land $CO_2$ flux from simulations with and without LUC, then the diagnosed LUC emissions depend on how simulations are performed. It is correct, that if concentration-driven simulations were to be used the diagnosed LUC emissions would have been higher and closer to Houghton (2008) estimates. We will clarify this.**

Discussion section

As commented above, simulations without LUC inevitably lead to lower $CO_2$ concentration and weaker $CO_2$ fertilization effect. I think this can be a "noise" when evaluating LUC emission/impacts. Specifically, in Fig.4(b), NPP in "without LUC" simulation are generally lower than "with LUC", but it is difficult to identify the reason of the difference, because the NPP difference can be affected by $CO_2$ fertilization, increased GPP by crops, and vegetation regrowth. I hope the authors to make a few discussions about the configuration settings for evaluating LUC impacts. I believe such information will be helpful when making simulation designs in the coming CMIP.

**We do not think that, when evaluating LUC emissions as the difference between atmosphere-land $CO_2$ flux in emissions-driven simulations with and without LUC, the lower $CO_2$ concentration in simulations without LUC can be considered as noise. In fact, lower $CO_2$ concentration in simulations without LUC, is expected and it is systematic. As re-**

**viewer # 3 suggested, our simulations could have been performed for the concentration-driven case. In that case, the diagnosed LUC emissions would have been higher and closer to Houghton (2008) estimates (as we mentioned above) and the difference in the rate of increase of NPP in simulations with and without LUC would have been solely due to differences in land cover. However, we would not have been able to use the fourth criterion, i.e. the amplitude of the annual $CO_2$ cycle and its rate of increase, to assess our simulations.**

**We note reviewer #2's point and will mention in our revised manuscript what the results would have been had we use concentration-driven simulations. However, we feel that while concentration-driven simulations make interpretation of results easier, emissions-driven simulations are more appropriate for our context. The real-world system is, of course, emissions-driven.**

In Fig.2, $\gamma_d = 0.25$ simulations display moderate land carbon sink among CMIP5-ESMs. I think this result is reasonable because most CMIP5-ESMs may not consider down-regulation mechanism; Fig.9 also supports the choice of the parameter value. However, the historical simulations with $\gamma_d = 0.25$ did not do a good job for reproducing land carbon uptake (Fig. 4). Although you discussed on this in the text, I suppose we have two more things to discuss. The first is the additional carbon uptake by vegetation regrowth. Although the regrowth mechanisms in the model are presented on p10-11, I'm not sure if the modeling was appropriate or not. If we can expect more carbon gain by vegetation regrowth, simulations with $\gamma_d = 0.25$ may work better. The second is the parameter value of humification factor. If you choose more moderate value for the humification factor (or modify the fractions of deforested/removed biomass that goes into fast/slow pools), soil carbon mass displayed in Fig. 5c will push up toward positive, and this treatment will also make the simulation with $\gamma_d = 0.25$ more realistic. I hope to see some discussions on these two points.

**This comment is somewhat unclear. The terrestrial ecosystem model used in our Earth system model grows vegetation in response to environmental conditions including atmospheric $CO_2$ concentration. Once the model reaches equilibrium, e.g. for environmental conditions corresponding to 1850, then a change in climate and/or atmospheric $CO_2$ concentration will make the model lose or gain carbon. Since $CO_2$ increases over the historical period then in a globally-averaged sense the model gains carbon creating the land carbon sink. We are unsure what "other" mechanisms can be used to grow vegetation.**

**As mentioned earlier, even if other models do not incorporate down-regulation it's the rate of increase of NPP over the historical period that is relevant here.**

**The second part of this comment raises the dilemma our study illustrates. Yes, we can use**

**a moderate value of the humification factor between 0.1 and 0.45 and use a lower value of $\gamma_d$ but that would yield lower soil carbon loss due to anthropogenic LUC and the carbon uptake for decades of 1960s through 2000s will likely not compare well with observation-based estimates from Le Quere et al. (2015), as was the case for CanESM2.**

**The other dilemma we faced is that while $\gamma_d = 0.4$ yields the best possible comparison with observation-based determinants of the global carbon cycle and historical carbon budget the model now yields carbon uptake that is highest amongst all CMIP5 models. This does not indicate that CanESM4.2 simulation of the historical carbon budget is grossly incorrect, but does make us an outlier amongst CMIP5 models.**

About Title:

I'm thinking the key feature of this paper is constraining the historical carbon budget of the model from different angles. Of course, it is necessary for your model to choose an appropriate value for the down-regulation, but its parameterization looks somewhat specific to your model. My suggestion is to change the title to reflect "$CO_2$ fertilization", "LUC", and "historical carbon budget": I believe these are the main issues in the background and will have more meaningful messages for readers.

**This is a valid suggestion and we will consider how the manuscript title may be changed.**
* * *
Reviewer # 3

Overall comment: This is an interesting study and asks an important question: how can we constrain an emergent propoerty such as the global responsiveness to elevated $CO_2$ based on global transient observational records? The authors are careful to emphasize the contingent nature of their answers, and emphasize that such answers cannot be unambiguously identified by this approach due to the presence of large amounts of model and forcing uncertainty that determine the response. I would personally go further and ask whether it makes sense at all to try to "tune" emergent model properties to match transient data in such an explicit manner. The more widely-followed approach is to test model components at scales where process-level understanding can be gained, in hopes of removing some of the dependence on overall model behavior that may influence results, for example by comparing at FACE sites (e.g. the various FACE-MIP papers), or by systematically benchmarking multiple aspects of the model in order to better understand the structural control on emergent behaviors. So while I do see this paper as a valuable contribution, I also feel that, in the end, the answer to the problem posed in the title is that it very much depends on what is in the ESM itself, and so without understanding how accurate the model is across a wide range of predictions, it is impossible to know whether the specific answer inferred by the comparison

is informative of the real world or not.

> Thank you for your interesting view point. Yes, it is true that traditionally models are evaluated using the bottom-up approach where aspects of the model are compared with observations to assess its various process-based parameterizations. The Canadian terrestrial ecosystem model (CTEM), which is the terrestrial carbon cycle component of CanESM4.2, has indeed been evaluated at point (e.g. Arora and Boer, 2005; Melton et al., 2015), regional (e.g. Peng et al., 2014; Garnaud et al., 2014) and global (e.g. Arora and Boer, 2010; Melton and Arora, 2014) scales in a number of studies. In regards to the $CO_2$ fertilization effect, based on results from FACE and other studies that grew plants at ambient and elevated $CO_2$, Arora et al. (2009) obtained a value of $\gamma_d$ equivalent to about 0.46 for use in CTEM.

> Just like top-down inversion-based studies are complementary to bottom-up studies (e.g. those which measure forest stem growth rates) in determining spatial distribution of carbon sinks and sources, we believe that there is value in evaluating and "tuning" CTEM using a top-down approach, as in our study, against an emergent model property. Amongst model simulations performed for our study for $\gamma_d$ = 0.25, 0.4 and 0.55, the simulation with $\gamma_d$ = 0.4 yields the best comparison with observation-based estimates. Indeed our "best" $\gamma_d$ of 0.4 is broadly consistent with Arora et al. (2009) derived $\gamma_d$ of 0.46 based on FACE studies.

> The tuned value of $\gamma_d$ is indeed model-dependent and we do mention this on top of page 26 of the discussion paper. To place confidence in the model, however, we attempt to compare different aspects of the model with observation-based estimates. These include loss in the global soil carbon amount due to anthropogenic LUC and the amplitude of annual $CO_2$ cycle and its rate of increase over the historical period.

> Finally, while the $\gamma_d$ parameter is specific to our model what's more useful for other modellers and the community at large is the simulated rate of increase of NPP over the historical period (which we explicitly mention in our abstract). The rate of increase of NPP can be directly compared across different models.

page 5, lines 7-14: I don't see how, from the perspective of the terrestrial biosphere, the information content of the first three of these tests are different. So if the focus is just on the land, why include the total $CO_2$ growth rate at all since the answer is affected by the uncertainty in ocean and fossil fuel emissions?

> Ignoring, the $CO_2$ growth rate over the historical period, against which model simulations are assessed, is certainly possible but that will make the simulations concentration-driven instead of being emissions-driven (which is what we have used in our study). The caveat

[Figure]

**with concentration-driven simulations is that it wouldn't be possible to analyze and use the amplitude of the annual $CO_2$ cycle and its rate of increase over the historical period to evaluate the model. Concentration-driven simulations either ignore the annual cycle of $CO_2$ (our specified-$CO_2$ case) or use a specified amplitude of the $CO_2$ annual cycle (our relaxed-$CO_2$ case). We do see value in comparing simulated and observed amplitude of the annual $CO_2$ cycle and its rate of increase over the historical period.**

**The information in cumulative atmosphere-land $CO_2$ flux for the period 1850-2005 and atmosphere-land $CO_2$ flux for the decades of 1960s through 2000s is actually different. This is shown in Figure 4 where the 1850-2005 cumulative atmosphere-land $CO_2$ flux for both CanESM2 and CanESM4.2 ($\gamma_d = 0.40$) lies within the uncertainty range of $-11\pm47$ PgC, but CanESM4.2 yields much better agreement with atmosphere-land $CO_2$ fluxes for the decades of 1960s through 2000s.**

page 14, discussion on "relaxed $CO_2$" approach. This seems to be a side point that isn't fully explained here, and I suggest either going into a bit more detail of what you mean (with figures or schematics) or else delete. Is the point that when you run it with relaxed $CO_2$, you are able to assess whether or or the model is in equilibrium? Or is the point that the 3D structure and seasonal variation of the $CO_2$ matters from a radiative perspective and therefore leads to a different baseline climate than in the specified $CO_2$ case?

**The 3D structure and seasonal variation of $CO_2$ will have some radiative implications but the effects will be second order since $CO_2$ is a fairly well-mixed greenhouse gas. More importantly, nonlinearity in the atmosphere-surface exchange of $CO_2$ means that the geographical structure and a seasonal cycle allowed in the relaxed $CO_2$ approach will produce a different atmosphere-surface $CO_2$ exchange than the specified $CO_2$ configuration (the relaxed being much more similar to the "free" $CO_2$ case). The "relaxed" $CO_2$ configuration is in fact the "free" $CO_2$ configuration with only the addition of a strong relaxation on the global-mean value of surface $CO_2$ to some specified reference value. Since all sources and sinks of $CO_2$ occur in the lowest model layer only, the reference value of $CO_2$ constrains the model but in a much more natural manner than specifying a uniform value of $CO_2$ everywhere. The text will be modified to clarify this point. The benefits of this approach have been described in the last paragraph of section 2.2.3.**

table 1: It might be useful to add a row here with the purpose of each scenario.

**We will make this modification.**

figure 1: Why is this functional form of the downregulaiton factor chosen? Assuming that the downregulation is meant to capture progressive nutrient limitation, it doesn't actually seem very progressive–the initial slope is quite high and then lessens at higher $CO_2$, but wouldn't one expect a priori that nutient limitations ought to become stronger only at higher $CO_2$ levels? Secondly, I can imagine that part of this phase space in this figure would be effectively excluded in that it would actually cause GPP to decrease under elevated $CO_2$, but it isn't apparent from the figure where that boundary would occur. have you performed sufficient sensitivity studies to identify where that transition is? Thirdly, the minimal downregulation case is quite close to the standard model, why was that chosen?

**Thank you for another good question. As explained in Arora et al. (2009), the functional form of the down-regulation factor derives from the fact that earlier simpler models of net or gross primary productivity (NPP or GPP) expressed it as a logarithmic function of atmospheric $CO_2$ concentration (e.g. Cao et al., 2001; Alexandrov and Oikawa, 2002).**

$$G(t) = G_0\left(1 + \gamma_p \ln\left(\frac{C(t)}{C_0}\right)\right) \tag{1}$$

**where GPP at any given time, $G(t)$, is a function of its initial value $G_0$, $CO_2$ concentration at time $t$, $C(t)$, and its initial value $C_0$. The rate of increase of GPP is determined by the parameter $\gamma_p$.**

**The ratio of GPP in two different versions of a model in which they increase at different rates ($\gamma_p$ and let's say $\gamma_d$) is given by**

$$\frac{1 + \gamma_d \ln\frac{C(t)}{C_0}}{1 + \gamma_p \ln\frac{C(t)}{C_0}} \tag{2}$$

**Equation (2) forms the basis for the functional form of down-regulation. For the case when $\gamma_d < \gamma_p$ the above ratio is less than 1 and its difference from 1 increases as $C(t)$ increase. In this sense the down-regulation is progressive. However, as reviewer # 3 notes the slope of down-regulation factor decreases. This second-order effect is a limitation of the formulation and we will make a note of this in revising our manuscript.**

**Reviewer # 3 raises a good point in regards to the boundary at which GPP may actually start to decrease with increasing $CO_2$. Although we have not derived the analytical equations that would allow to find where this boundary occurs, the model does not show any indication of decreasing GPP at least up until atmospheric $CO_2$ concentration of around**

[Figure]

**1000 ppm (as in the RCP 8.5 scenario) (see Arora and Boer, 2014). Although not relevant for this manuscript this comment provides us the reason to derive those analytical equations.**

**Finally, the values of $\gamma_d$ chosen are equal to 0.40±0.15. While the minimal downregulation case ($\gamma_d = 0.55$) appears close to the standard model ($\gamma_d = 0.4$) in Figure 1 this isn't in the case for the results obtained (see e.g. Figures 4a, 5a, 5c, and 7) because of the non-linear behaviour of the system.**

page 28, last paragraph. Implicit in this argument seems to be the idea that the degree of historical growth and the response of the terrestrial biosphere over the historical period ought to be informative of an idealized 1%/yr forcing. But to the extent that downregulation is progressively driven by nutrient limitations, it ought to be expressed differently based on the rate at which $CO_2$ increases. So it may be just as informative to consider an extremely rapid $CO_2$ increase even if not at global scale, as in a FACE experiment, as it is to consider the slower-than 1%/yr forcing that has been applied globally over the historical period.

**It wasn't our intent to imply that the 1% per year increasing $CO_2$ simulation is in any way indicative of the model response over the historical period, or vice-versa. The reference to Figure 2 again in this last paragraph of the manuscript was merely to mention that the value of $\gamma_d = 0.4$ that gives best comparison against observation-based determinants of the historical carbon budget makes CanESM4.2 an outlier amongst CMIP5 models. We will clarify this.**

————————————————————————————

**References**

Alexandrov, G., and T. Oikawa: TsuBiMo: a biosphere model of the $CO_2$ fertilization effect, 750 Clim. Res., 19, 265-270, 2002.

Arora, V. K. and Boer, G. J.: Uncertainties in the 20th century carbon budget associated with land use change, Glob. Change Biol., 16, 3327–3348, 2010.

Arora, V. K. and Boer, G. J.: A parameterization of leaf phenology for the terrestrial ecosystem component of climate models, Glob. Change Biol., 11, 39–59, doi:10.1111/j.1365-2486.2004.00890.x, 2005.

Arora, V. K. and Boer, G. J.: Terrestrial ecosystems response to future changes in climate and atmospheric $CO_2$ concentration, Biogeosciences, 11, 4157-4171, doi:10.5194/bg-11-4157-2014, 2014.

Arora, V. K., Boer, G. J., Christian, J. R., Curry, C. L., Denman, K. L., Zahariev, K., Flato, G. M., Scinocca, J. F., Merryfield, W. J., and Lee, W. G.: The effect of terrestrial photosynthesis down-regulation on the

20th century carbon budget simulated with the CCCma Earth System Model, J. Climate, 22, 6066–6088, 2009.

Cao, M., Q. Zhang, and H.H. Shugart: Dynamic responses of African ecosystem carbon cycling to climate change, Clim. Res., 17, 183-193, 2001.

Garnaud, C., L. Sushama, V. K. Arora: The effect of driving climate data on the simulated terrestrial carbon pools and fluxes over North America, International Journal of Climatology 34 (4), 1098-1110, 2014.

Houghton, R. A.: Carbon flux to the atmosphere from land-use changes: 1850–2005, in: TRENDS: a Compendium of Data on Global Change, Carbon Dioxide Information Analysis Center, Oak Ridge National Laboratory, U.S. Department of Energy, Oak Ridge, Tenn., USA, 2008.

Le Quéré, C., Moriarty, R., Andrew, R. M., Peters, G. P., Ciais, P., Friedlingstein, P., Jones, S. D., Sitch, S., Tans, P., Arneth, A., Boden, T. A., Bopp, L., Bozec, Y., Canadell, J. G., Chini, L. P., Chevallier, F., Cosca, C. E., Harris, I., Hoppema, M., Houghton, R. A., House, J. I., Jain, A. K., Johannessen, T., Kato, E., Keeling, R. F., Kitidis, V., Klein Goldewijk, K., Koven, C., Landa, C. S., Landschützer, P., Lenton, A., Lima, I. D., Marland, G., Mathis, J. T., Metzl, N., Nojiri, Y., Olsen, A., Ono, T., Peng, S., Peters, W., Pfeil, B., Poulter, B., Raupach, M. R., Reg nier, P., Rödenbeck, C., Saito, S., Salisbury, J. E., Schuster, U., Schwinger, J., Séférian, R.,Segschneider, J., Steinhoff, T., Stocker, B. D., Sutton, A. J., Takahashi, T., Tilbrook, B., van der Werf, G. R., Viovy, N., Wang, Y.-P., Wanninkhof, R., Wiltshire, A., and Zeng, N.: Global carbon budget 2014, Earth Syst. Sci. Data, 7, 47–85, doi:10.5194/essd-7-47-2015, 2015.

Melton, J. R. and Arora, V. K.: Sub-grid scale representation of vegetation in global land surface schemes: implications for estimation of the terrestrial carbon sink, Biogeosciences, 11, 1021-1036, doi:10.5194/bg-11-1021-2014, 2014.

Melton, J. R., Shrestha, R. K., and Arora, V. K.: The influence of soils on heterotrophic respiration exerts a strong control on net ecosystem productivity in seasonally dry Amazonian forests, Biogeosciences, 12, 1151-1168, doi:10.5194/bg-12-1151-2015, 2015.

Peng, Y., Arora, V. K., Kurz, W. A., Hember, R. A., Hawkins, B. J., Fyfe, J. C., and Werner, A. T.: Climate and atmospheric drivers of historical terrestrial carbon uptake in the province of British Columbia, Canada, Biogeosciences, 11, 635-649, doi:10.5194/bg-11-635-2014, 2014.

Schimel, D., Stephens, B. B., and Fisher, J. B.: Effect of increasing $CO_2$ on the terrestrial carbon cycle, Proceedings of the National Academy of Science U.S.A., 112, 436–441, doi:10.1073/pnas.1407302112, 2015.